# FAST: Federated Average with Snapshot Unleashes Arbitrary Client Participation

## Abstract

Federated Learning (FL) provides a flexible distributed platform where numerous clients with high degrees of heterogeneity in data and system can collaborate to learn a model jointly. Previous research has shown that FL is effective in handling diverse data but often assumes idealized conditions. Specifically, client participation is often simplified in these studies, while real-world factors make it difficult to predict or design individual client participation. This complexity often diverges from the ideal client participation assumption, rendering an unknown pattern of client participation, referred to as *arbitrary client participation (ACP)*. Hence, it is an important open problem to explore the impact of client participation and find a lightweight mechanism to enable ACP in FL. In this paper, we first empirically investigate the influence of client participation on FL, revealing that FL algorithms are significantly impacted by ACP. To alleviate the influence, we propose a lightweight solution, Federated Average with Snapshot (FAST), to unleash almost ACP for FL. It can seamlessly integrate with other classic FL algorithms. Specifically, FAST enforces the clients to take a snapshot once in a while and facilitates ACP for the majority of the training process. We show the convergence rates of FAST in non-convex and strongly-convex cases, which match the rates with those in ideal client participation. Furthermore, we empirically introduce an adaptive strategy for dynamically configuring the snapshot frequency, tailored to accommodate diverse FL systems. Our extensive numerical results demonstrate that our FAST attains significant improvements under the conditions of ACP and highly heterogeneous data.

## 1 Introduction

Federated Learning (FL) stands out as an emerging distributed machine learning framework where a large number of clients (i.e., computing nodes or devices) collaborate together to train a machine learning model under the coordination of a central server (McMahan et al., 2017; Kairouz et al., 2021). FL establishes itself as a powerful and flexible distributed platform, fostering collaboration among diverse clients characterized by substantial heterogeneity in data and system while preserving the privacy of raw data residing within each client. Hence, previous research endeavors have yielded a spectrum of efficient algorithms capable of achieving optimal convergence rates in theory and delivering great performance in some practical cases, in the presence of varying degrees of data heterogeneity (Kairouz et al., 2021; Zhao et al., 2018; Li et al., 2019; Karimireddy et al., 2020; Yang et al., 2020; Wang et al., 2021).

Nevertheless, realizing these favorable outcomes often hinges on the assumption of an ideal system condition (i.e., ideal client participation). Specifically, most FL algorithms presume that client participation can be fully known, controlled, predicted, or tracked. For example, some works assume partial client participation, where participation follows a known or controllable random process, such as ergodic, mixing, or independent processes (McMahan et al., 2017; Acar et al., 2021; Cho et al., 2023). Others assume that each client participates at least once within certain rounds (Yang et al., 2022b; Gu et al., 2021; Yan et al., 2024). In practice, however, each client's participation is **highly dynamic, unknown and unpredictable** (Bonawitz et al., 2019; Soltani et al., 2022) since clients frequently exhibit a spectrum of heterogeneous and dynamically shifting attributes, including computational power, communication capacity, and availability (Kairouz et al., 2021; Bonawitz et al., 2019; Yang et al., 2021). These variations stem from the unique characteristics of each individual

client and the dynamics of distributed learning systems. The intricacies of client participation, marked by their dynamic, unknown, and unpredictable nature, make it challenging and even impossible to ascertain a priori beforehand. Moreover, in some FL systems, such as cross-device FL, tracking client participation is either infeasible or not permitted Kairouz et al. (2021). We refer to this complex and unpredictable pattern as ***arbitrary client participation (ACP)***, reflecting its dependence on various system factors and the absence of explicit client tracking. Clearly, it leaves a substantial gap between algorithmic designs built on the premise of ideal client participation and the real-world applications of FL in the face of ACP. On the other hand, without any conditions on client participation, a constant error arises for ACP as identified by the lower bound Cho et al. (2022); Wang et al. (2020); Yang et al. (2022b), implying that no algorithm can achieve stationary point convergence in such case. This observation motivates us to pose the following fundamental question:

> **(Q1)**: Is it possible to design a lightweight mechanism for FL that can accommodate arbitrary client participation with theoretical guarantees?

In this paper, we show an affirmative answer to this question by proposing a new client participation mechanism for FL, denoted as Federated Averaging with SnapshoT (FAST). In contrast to most FL algorithms that necessitate ideal client participation in each communication round, FAST imposes a minimal requirement for client participation by intermittently implementing a snapshot step. This approach significantly diminishes the requirement for individual client participation, enabling ACP for the majority of the training process. We highlight our contributions as follows:

- Through extensive experiments, we revealed that the mismatch between ideal client participation in algorithm design and arbitrary client participation in practice leads to severe performance degradation, especially in highly heterogeneous data scenarios. These phenomena are universal and extend beyond specific algorithms, as observed across multiple FL algorithms.

- To address this issue, we introduce FAST, a lightweight FL framework that requires only intermittent snapshot steps, enforcing fully random client participation during these steps while accommodating arbitrary client participation within the system at all other times. This requirement applies to the client cohort rather than individual clients, allowing the participating group to be statistically representative. This is a milder condition compared to existing works (see Table 1), as it eliminates the need to track each client individually.

- Theoretically, we demonstrate that, under mild conditions, FAST can achieve a convergence rate of $\mathcal{O}(1/\sqrt{mRK})$ for non-convex functions and $\widetilde{\mathcal{O}}(1/R)$ for strongly-convex functions, where $R$ is the number of communication rounds, $K$ is the number of local steps, and $m$ is the number of participated clients. These rates can match the rates of that with ideal client participation.

- Empirically, we further propose an adaptive strategy designed to adjust the frequency of the snapshot step dynamically, and we show that our FAST framework can seamlessly integrate with other classic FL algorithms. Also, our extensive experiments show its effectiveness.

Table 1: Comparison of Client Participation in FL and Convergence Rate in Non-convex Functions.

| Algorithm | Participation Condition | Client Tracking | Convergence Rate |
|---|---|---|---|
| MIFA (Gu et al., 2021) | Bounded inactive rounds | ✓ | $\mathcal{O}(\frac{1}{\sqrt{mKR}})$ |
| AFL (Yang et al., 2022b) | Bounded inactive rounds | ✓ | $\mathcal{O}(\frac{1}{\sqrt{mKR}})$ |
| FedAU (Wang & Ji, 2023) | Every client participates | ✓ | $\mathcal{O}(\frac{1}{\sqrt{mKR}})$ |
| FedAmplify (Wang & Ji, 2022) | Regularized, mixing, independent process | ✗ | $\mathcal{O}(\frac{1}{\sqrt{mKR}})$ |
| FedAvg (McMahan et al., 2017) | Uniform participation in every round | ✗ | $\mathcal{O}(\frac{1}{\sqrt{mKR}})$ |
| FAST (our work) | Uniform participation occasionally | ✗ | $\mathcal{O}(\frac{1}{\sqrt{mKR}})$ |
| Lower Bounds (Cho et al., 2022; Wang et al., 2020; Yang et al., 2022b) | No assumptions | - | $\Omega(1)$ |

## 2 RELATED WORK

**Ideal Client Participation: full client participation and uniformly random client participation.** In FL, client participation can be seen as a proxy for system heterogeneity. Due to the inherent complexity of real-world FL systems, explicitly modeling client participation proves challenging (Bonawitz et al., 2019; Yang et al., 2021). Most existing FL algorithms often make an assumption about ideal client participation, typically relying on either full client participation (Gorbunov et al., 2021; Haddadpour et al., 2019; Lin et al., 2018; Wang & Joshi, 2019; 2021; Yu et al., 2019) or

uniformly random client participation (McMahan et al., 2017; Li et al., 2019; Karimireddy et al., 2020; Yang et al., 2020; Wu et al., 2023; Zhang et al., 2023; Wang et al., 2023; Liu et al., 2021; Jhunjhunwala et al., 2022; Grudzień et al., 2023). This assumption requires that the server can force all clients or at least uniformly and randomly sample a subset of clients to participate in each communication round. However, it is crucial to acknowledge that each client in FL is not entirely under the server's control. While the server may sample a client for a specific round, the client is highly likely not to participate due to various system factors such as drop-out, communication congestion, and other unpredictable factors (Kairouz et al., 2021; Yang et al., 2021). It is worth noting that the server can invest additional resources to enforce uniform client participation, such as sampling more clients and extending the waiting time in each round. Yet, this approach leads to prolonged training times due to significant communication and computation overhead (Zhou et al., 2022). As shown in (Luo et al., 2022), enforcing uniform client participation in every round by the server results in slow wall-clock time for FL training.

**Controllable Client Participation.** In addition to uniform client participation, another approach in the field involves modeling client participation as a controllable random process. One line of works utilizes predefined patterns or probabilities as the model of client participation (Chen et al., 2022; Yang et al., 2022b; Fraboni et al., 2021; Ruan et al., 2021; Gu et al., 2021; Avdiukhin & Kasiviswanathan, 2021; Wang & Ji, 2022; Koloskova et al., 2022). The main idea is to allow asynchronous communication or fixed participation patterns (e.g., given probability) for clients to participate flexibly in training. However, existing works in this area often require extra assumptions, such as bounded delay and extra memory (Yang et al., 2022b; Ruan et al., 2021; Gu et al., 2021; Koloskova et al., 2022) and identical computation rate (Avdiukhin & Kasiviswanathan, 2021). Moreover, several works explore some unique scenarios of client participation. For instance, (Chen et al., 2022) introduced a novel client subsampling scheme considering the importance of updates, relying solely on the norm of the update. The studies by (Malinovsky et al., 2023) and (Cho et al., 2023) investigated cyclic client participation. (Wang & Ji, 2022) provided a unified analysis for various client participation, including regularized, ergodic, independent, and mixing participation. The implicit assumption in these studies is that client participation is either known, largely controllable or adheres to predefined patterns. It is also noteworthy to mention a related work (Wang & Ji, 2023), wherein the estimated probability of each client's participation was used for a re-weighting process under unknown participation statistics. However, estimating such probabilities can be challenging in practical scenarios, such as cross-device FL (Kairouz et al., 2021).

Each of these approaches contributes to the diverse client participation strategies employed in FL. However, these strategies often necessitate adherence to specific patterns, which may not align seamlessly with practical FL scenarios characterized by *highly dynamic, unknown and unpredictable* nature. In this paper, we introduce a more general and practical approach, referred to as *arbitrary client participation (ACP)*. This implies that we do not impose any assumptions on client participation for the majority of the training round. Our aim is to offer a flexible and realistic framework that accommodates various client participation scenarios in real-world FL applications.

**Comparison of Related Work.** We compare some related work about ACP in Table 1. Except for the difference in participation patterns and convergence rate, there are still some important points that we need to compare. For FedAmplify (Wang & Ji, 2022), it can achieve the convergence rate of $\mathcal{O}(\frac{1}{\sqrt{mKR}})$ only in some ideal cases (see Sec. 5 in (Wang & Ji, 2022)), and the server requires participation frequency for each client. For MIFA (Gu et al., 2021), each client needs to participate in training at least once in the one-time window. For Anarchic Federated Learning (AFL) (Yang et al., 2022b), the server needs to identify and store local models, and each client needs to participate in training at least once in the one-time window. In contrast, our FAST framework has no extra assumptions for client participation and can achieve the ideal convergence rate. In addition, regular FAST does not have any demand to store extra information.

## 3 THE IMPACT OF CLIENT PARTICIPATION IN FL

In this section, our goal is to investigate the impact of client participation on FL performance. We first introduce the fundamental formulation and the standard FedAvg. Subsequently, we examine FedAvg's performance across various client participation scenarios and show the adverse effects of different ACP. This highlights the gap between current algorithm designs and practical FL systems, thus motivating us to develop a new framework to accommodate ACP for FL.

## 3.1 Federated Learning and Federated Averaging

**Problem Formulation.** In one FL system with $M$ clients, the goal is to minimize the objective function, which can be formulated as follows:

$$\min_{\boldsymbol{x} \in \mathbb{R}^d} F(\boldsymbol{x}) := \frac{1}{M} \sum_{i=1}^{M} F_i(\boldsymbol{x}), \tag{1}$$

where $\boldsymbol{x} \in \mathbb{R}^d$ is a $d$-dimension model parameter, $M$ is the total number of clients, and $F_i(\boldsymbol{x}) := \frac{1}{|S_i|} \sum_{\xi \in D_i} F(\boldsymbol{x}, \xi), \forall i \in [M]$ is the local loss function associate with local dataset $D_i$ that is IID sampled from one underlying distribution $P_i$. One of the critical features of FL is that each client has a subtly different local data distribution, i.e., $P_i \neq P_j$ if $i \neq j$. This leads to heterogeneous (or Non-IID) data in the FL system, causing model drift and non-trivial performance degradation (Kairouz et al., 2021; Wang et al., 2021).

**FedAvg Algorithm.** The Federated Average (FedAvg) algorithm (McMahan et al., 2017) stands as the pioneering exemplar algorithm for FL, inspiring numerous followup algorithms. Most of the FL algorithms follow the typical parameter-server architecture. In each communication round $r \in [R]$, the server first selects a subset of clients to participate and broadcasts the current global model $\boldsymbol{x}_r$ to each client. Upon receiving the global model, each participating client locally optimizes the loss function for some local steps using the local dataset without communication. For example, FedAvg takes $K$ local steps using the vanilla stochastic gradient descent method. That is, $\boldsymbol{x}_{r,k+1}^i = \boldsymbol{x}_{r,k}^i - \eta_c \nabla F_i(\boldsymbol{x}_{r,k}^i, \xi_{r,k}^i), k \in \{0, \cdots, K-1\}$ starting from $\boldsymbol{x}_{r,0}^i = \boldsymbol{x}_r$ where $\xi_{r,k}^i \sim D_i$. After the local computation, the client sends the model update $\boldsymbol{x}_r^i = \boldsymbol{x}_{r,K}^i$ to the server. At the server side, the server updates the global model by aggregating all the returned local model, i.e., $\boldsymbol{x}_{r+1} = \frac{1}{|S_r|} \sum_{i \in S_r} \boldsymbol{x}_r^i$ where $S_r$ is the set of participated clients in the $r$-th round. Then, the next training round begins.

Undoubtedly, client participation, denoted as the set $S_r$, stands as a pivotal factor influencing the performance of FL models. While the majority of works in FL concentrate on mitigating data heterogeneity, the implications of client participation remain largely under-explored. To ensure convergence guarantees in FL algorithms, specific conditions must be imposed on client participation. Essentially, these algorithms necessitate a regulated form of client participation, such as participation through uniformly random sampling or a predetermined probability distribution, as detailed in Sec.2.

However, in real-world FL systems, client participation is inherently dynamic, prone to changes in each round (Bonawitz et al., 2019; Yang et al., 2021). Even if the server employs an ideal sampling way, like uniformly random sampling, actual client participation remains unknown and largely uncontrollable. We term this as ***arbitrary client participation***, signifying that $S_r$ includes any sampling from the whole client set $[M]$, thereby incorporating a diverse array of participation schemes. This process is determined by various inherent system factors, such as client failures and status changes (Bonawitz et al., 2019; Yang et al., 2021). Hence, there exists a conflict between current algorithm designs with ideal client participation and practical FL systems with ACP. This motivates us to explore the impact of different client participations on the FL algorithms' performance.

## 3.2 The Impact of Client Participation in FL

**Simulation of Arbitrary Client Participation.** We delve into FedAvg's performance across four client participations characterized by distinct distributions: uniform, Beta, Gamma, and Weibull. Uniform client participation entails the random client selection from the entire client set, which is an assumption representing the ideal scenario in current FL algorithms. The Beta distribution is commonly employed to model events constrained within an interval. The Gamma distribution finds application in characterizing the frequency of a sequence of events associated with time or distance, while the Weibull distribution is widely utilized in reliability or survival analysis (Lai et al., 2006). In FL, the server often receives returns from clients within a given time window. Hence, it is reasonable to use uniform distribution as a baseline for ideal client participation. The latter three distributions are utilized to approximate different real-world scenarios, serving as representatives of ACP.

It is important to emphasize that our primary goal is not to precisely model client participation in FL but to explore the impact of different potential client participation scenarios. Also, we aim to highlight the adverse effects resulting from the mismatch between the ideal client participation used in the current algorithm design and ACP observed in practical FL.

**Experiment Settings.** We perform extensive experiments on Fashion-MNIST (Xiao et al., 2017) and CIFAR-10 (Krizhevsky et al., 2009), considering various Non-IID degrees and utilizing these four distributions to simulate different client participation. As shown in Table 2, we scrutinize the model performance using FedAvg. For each case, we record the last five results and report the mean and standard deviation of test accuracy. Due to the space limit, we only show key findings and delegate the detailed settings and results for other datasets and algorithms to Sec. 5 and Appendix B.2.

Table 2: Test Accuracy Comparison of FedAvg

| Participation \ $\alpha$ | Fashion-MNIST | | | | | CIFAR-10 | | |
|---|---|---|---|---|---|---|---|---|
| | 0.05 | 0.1 | 0.3 | 0.5 | 1.0 | 0.1 | 0.5 | 1.0 |
| Uniform | **84.10%±2.4** | **86.85%±1.9** | **89.39%±0.7** | **91.39%±0.3** | **92.21%±0.3** | **80.18%±0.6** | **80.49%±0.4** | **80.83%±0.7** |
| Beta | 74.84%±1.2 | 79.89%±4.0 | 86.40%±1.1 | 88.74%±0.4 | 89.43%±0.1 | 68.30%±0.9 | 72.27%±0.4 | 73.32%±0.6 |
| Gamma | 66.65%±4.7 | 81.81%±1.8 | 88.41%±0.5 | 87.79%±0.4 | 89.44%±0.2 | 70.90%±0.8 | 73.20%±0.4 | 73.04%±0.3 |
| Weibull | 73.15%±5.1 | 78.78%±1.6 | 88.80%±0.4 | 89.20%±0.6 | 89.53%±0.2 | 71.74%±0.7 | 73.21%±0.7 | 73.75%±0.3 |

$^*$ The details of this table are introduced in Sec. 5-Note.

**Observations.** We have three key observations. First, the performance of FedAvg is significantly influenced by client participation. As shown in Table 2, the model accuracy varies across different client participation cases, with uniform participation yielding the best performance among these four cases. This performance difference is substantial, ranging from $3\%$ to $18\%$. These results align with practical FL simulations, where uncontrolled client participation induced by system heterogeneity leads to non-trivial model performance degradation (Yang et al., 2021). Second, this performance degradation strongly correlates with the degree of Non-IID data. In our setting, we adopt the common approach of generating Non-IID data using the Dirichlet distribution (Acar et al., 2021), with the parameter $\alpha$ controlling the Non-IID degree. A smaller $\alpha$ corresponds to a higher Non-IID degree. For datasets with a higher degree of Non-IID data (smaller $\alpha$), the model accuracy gap between uniform and other cases becomes more pronounced. For instance, on the Fashion-MNIST dataset, the model behaves similarly for different client participation cases with less Non-IID data (i.e., $\alpha = 1$). However, as the degree of Non-IID gets higher, such as $\alpha = 0.05$, the accuracy gap between uniform and other participation cases could be as large as $18\%$. Third, the performance degradation for ACP (in the latter three cases) is a universal phenomenon. This extends beyond FedAvg, as evidenced by consistent observations across other FL algorithms such as FedProx and FedAvgM.

It is essential to note that occasional enforcement of uniform client participation in FL is feasible. For instance, the server can sample a larger number of clients and allocate sufficient time for each communication round, allowing ample clients to complete local computations. However, this strategy inevitably demands more resources and significantly extends the training time due to longer waiting time. Therefore, it becomes unrealistic to enforce uniform client participation in every round. On the other hand, without imposing any constraints on client participation, FedAvg is theoretically incapable of asymptotically converging to a stationary point (Yang et al., 2022b;a) and experiences non-trivial performance degradation in practice, as shown above. This realization motivates us to develop a lightweight client participation mechanism. This mechanism aims to achieve performance similar to that of uniform participation while imposing fewer constraints on FL systems.

## 4 FEDERATED AVERAGE WITH SNAPSHOT (FAST)

In this section, we first introduce a lightweight client participation mechanism, denoted as Federated Average with SnapshoT (FAST). Then, we provide the convergence analysis in non-convex and strongly-convex cases. In addition, to eliminate the requirement to set snapshot frequency in advance, we empirically propose a strategy to adjust the snapshot frequency for our FAST adaptively.

### 4.1 ALGORITHM DESCRIPTION

As illustrated in Algorithm 1, we introduce a lightweight and practical client participation mechanism for FL. In each communication round $r \in [R]$, we design two options for client participation. If $r\%I == 0$, the server takes a snapshot step that requires to enforce a round of uniform client participation denoted as client set $S_r^u$ with cardinality $m$ for that round (Lines 3-4), where $I$ is a hyper-parameter to control the frequency of the snapshot step. Otherwise, the server does not put any constraints and can accommodate any system heterogeneity by allowing ACP denoted as set $S_r^a$ with cardinality $n$ (Lines 5-6). On the client side, each participating client takes $K$ Stochastic Gradient

---

**Algorithm 1** Federated Average with Snapshot (FAST)

1: Initialize model parameter $\boldsymbol{x}_0$, learning rate $\eta_c$, the number of local update steps $K$, communication rounds $R$, snapshot step interval $I$ (or probability $q$).
2: **for** $r = 0, \ldots, R-1$ **do**
3:     If $r\%I == 0$ (with probability $q = 1/I$):              ▶ Snapshot
4:        The server enforces *uniformly* random clients $\mathcal{S}_r = \mathcal{S}_r^u$ ($|\mathcal{S}_r^u| = m$) to participate
5:     Otherwise:                                             ▶ Arbitrary
6:        The server allows *arbitrarily* random clients $\mathcal{S}_r = \mathcal{S}_r^a$ ($|\mathcal{S}_r^a| = n$) to participate
7:     Each client $i \in \mathcal{S}_r$ computes in parallel:
8:        Local update: $\boldsymbol{x}_{r,k+1}^i = \boldsymbol{x}_{r,k}^i - \eta_c \nabla F_i(\boldsymbol{x}_{r,k}^i, \xi_{r,k}^i), k \in [K]$.
9:        Send $\boldsymbol{x}_r^i = \boldsymbol{x}_{r,k+1}^i$ to the server.
10:    Server aggregation: $\boldsymbol{x}_{r+1} = \frac{1}{|\mathcal{S}_r|} \sum\limits_{i \in \mathcal{S}_r} \boldsymbol{x}_r^i$.
11: **end for**

---

Descent (SGD) steps and sends the returns back to the server (Lines 7-9), mirroring the procedure in the FedAvg algorithm. Subsequently, after local computations, the server aggregates all the returns and updates the global model (Line 10). Additionally, we present a probabilistic perspective. In each round, there exists a probability $q$ of enforcing snapshots and a complementary probability of $1-q$ to permit ACP. Here $q = 1/I$ can be interpreted as the snapshot probability or frequency.

In general, the uniqueness of FAST is utilizing a snapshot step every $I$ rounds by enforcing a round of uniform client participation. The trade-offs of the snapshot are discussed as follows: 1) Resources. Although uniform client participation is an ideal situation in FL, it can still be achieved in practice by using some strategies. For instance, the server can initially sample $1.3 \times m$ clients and extend the waiting period Bonawitz et al. (2019). This approach would make uniformly random client participation hold statistically, and mirrors practical FL simulations, such as 11.6% dropout rate and an optimal waiting time Yang et al. (2021). Hence, enforcing uniform client participation is practical in reality. Unfortunately, this approach to achieve uniform participation consumes more resources, such as time and computation. However, in FAST, snapshots just occupy a small portion of entire training rounds, so FAST can save resources compared to completely uniform participation in other FL algorithms. 2) Benefits. By the snapshot, our FAST can simultaneously enjoy the optimal convergence rates as those with uniform client participation shown in Sec. 4.2 and achieve improved performance when compared with ACP shown in Sec. 5.

### 4.2 CONVERGENCE ANALYSIS

We first state several standard assumptions commonly used in our work and other optimization and FL works (Kairouz et al., 2021; Wang et al., 2021).

**Assumption 1 ($L$-Lipschitz Continuous Gradient)** *For any $\boldsymbol{x}$ and $\boldsymbol{y}$, there exists a constant $L > 0$ such that $\|\nabla F(\boldsymbol{x}) - \nabla F(\boldsymbol{y})\| \leq L\|\boldsymbol{x} - \boldsymbol{y}\|$ and $\|\nabla F_i(\boldsymbol{x}) - \nabla F_i(\boldsymbol{y})\| \leq L\|\boldsymbol{x} - \boldsymbol{y}\|$.*

**Assumption 2 (Unbiased Stochastic Gradients with Bounded Variance)** *The stochastic gradient calculated by the client or server is unbiased with bounded variance: $\mathbb{E}[\nabla F_i(\boldsymbol{x}, \xi)] = \nabla F_i(\boldsymbol{x})$ and $\mathbb{E}[\|\nabla F_i(\boldsymbol{x}, \xi) - \nabla F_i(\boldsymbol{x})\|^2] \leq \sigma^2$, where $\xi$ is the data sample.*

**Assumption 3 (Bounded Gradient Dissimilarity)** *For any $i \in [M]$, $\|\nabla F_i(\boldsymbol{x}) - \nabla F(\boldsymbol{x})\|^2 \leq \sigma_G^2$.*

Now, we are ready to offer FAST's convergence analysis for non-convex functions.

**Theorem 1** (Convergence of FAST for Non-convex Functions). *Under the Assumptions 1, 2 and 3, supposing that the probability $q \geq \frac{(2LK\eta_c - 1)G_2 + 2K^2\sigma_G^2}{G_1 + (2LK\eta_c - 1)G_2 - 2LK\eta_c G_3 + 2K^2\sigma_G^2}$ and the learning rate $\eta_c \leq \min\left\{\frac{1}{8LK}, \frac{nq + m(1-q)}{5mnLK}\right\}$, then the sequence $\{\boldsymbol{x}_r\}$ generated by FAST satisfies:*

$$\frac{1}{R}\sum_{r=1}^{R} \mathbb{E}\|\nabla F(\boldsymbol{x}_r)\|^2 \leq \underbrace{\frac{4\zeta}{KR\eta_c}}_{\text{Optimization Error}} + \underbrace{\frac{4(qn + (1-q)m)}{mn} L\eta_c \sigma^2}_{\text{Statistical Error}} + \underbrace{\left(120(1-q) + 60q\right)L^2 K^2 \eta_c^2 \sigma_G^2}_{\text{Heterogeneity Error}}, \quad (2)$$

*where $\zeta := F(\boldsymbol{x}_0) - F(\boldsymbol{x}^*)$, $\boldsymbol{x}^*$ is the optimal solution, and $G_{1-3}$ are defined in the Appendix A.2.*

With a proper learning rate, we have the following convergence rate for FAST:

**Corollary 1** *With $\eta_c = \mathcal{O}\left(\frac{\sqrt{mn}}{\sqrt{RK(nq+m(1-q))}}\right)$, the convergence rate of FAST is*

$$\frac{1}{R}\sum_{r=1}^{R}\mathbb{E}\|\nabla F(\boldsymbol{x}_r)\|^2 \leq \mathcal{O}\left(\sqrt{\frac{nq+m(1-q)}{nmKR}}\right) + \mathcal{O}\left(\frac{mnK}{(nq+(1-q)m)R}\right) \qquad (3)$$

The convergence error of FAST comprises three components: 1) the optimization error depending on the initial point $\boldsymbol{x}_0$, 2) the statistical error associated with stochastic gradient noise $\sigma$, and 3) the error arising from heterogeneous data and local updates in FL. Notably, the third error exhibits a quadratic relationship with the learning rate. Hence, the first two terms dominate when using a sufficiently small learning rate. With an appropriate learning rate, the convergence rate is $\mathcal{O}\left(\sqrt{\frac{nq+m(1-q)}{nmKR}}\right)$ for a suitably large round $R \geq \frac{(mnK)^3}{[nq+m(1-q)]^3}$. In the special case($m = n$), the convergence rate becomes:

**Corollary 2** *Supposing that $m = n$, FAST achieves convergence rate:*

$$\frac{1}{R}\sum_{r=1}^{R}\|\nabla F(\boldsymbol{x}_r)\|^2 \leq \mathcal{O}\left(\sqrt{\frac{1}{mRK}}\right). \qquad (4)$$

**Remark 1** In non-convex functions, this sublinear convergence rate shows the speedup in terms of clients' number $m$ and the local steps $K$, which matches the optimal convergence rate in FL with uniform client participation in every round (Karimireddy et al., 2020; Yang et al., 2020).

**Remark 2** It is worth pointing out that there does exist a requirement of the snapshot probability/frequency $q$ (or $I$). Specifically, it depends on data heterogeneity in the FL system: $q \geq \frac{(2LK\eta_c-1)G_2+2K^2\sigma_G^2}{G_1+(2LK\eta_c-1)G_2-2LK\eta_cG_3+2K^2\sigma_G^2} = \frac{1}{1+(G_1-2LK\eta_cG_3)/(2K^2\sigma_G^2+2LK\eta_cG_2-G_2)}$. For every heterogeneous data in FL, we can choose a proper $q$ such that it can converge at such an optimal rate. We list two special cases to show FAST's generalization. 1) $\sigma_G \to 0$. If data is IID among clients, then $q \geq 0$, meaning that we can always avoid using the snapshot step and set $q = 0$. This situation corresponds to traditional distributed learning where each client has access to a shared dataset or IID datasets. In such cases, the choice of which subset of clients participates is inconsequential, as the training data used remains statistically identical. 2) $\sigma_G \to \infty$. If data is extremely highly Non-IID, the lower bound of $q$ will approach $1$, requiring a high frequency of snapshots. In extreme cases, it might require uniform client participation in every round to guarantee convergence.

If we assume a strongly convex condition on the function, we can achieve a faster convergence rate.

**Assumption 4 (Strong Convexity)** *For any $\boldsymbol{x}$ and $\boldsymbol{y}$, $F_i$ is $\mu$-convex with a constant $\mu > 0$, if $F_i(\boldsymbol{y}) \geq F_i(\boldsymbol{x}) + \nabla F_i(\boldsymbol{x})^T(\boldsymbol{y} - \boldsymbol{x}) + \frac{\mu}{2}\|\boldsymbol{y} - \boldsymbol{x}\|^2, \forall i \in [M]$.*

**Theorem 2** (Convergence of FAST for Strongly Convex Functions). *Under the Assumptions 1,2,3 and 4, supposing that the learning rate $0 < \eta_c \leq \min\left\{\frac{1}{20mLK}, \frac{1}{20nLK}\right\}$ and the probability $q \geq 1 - \left(\frac{\mu K\eta_c-16L^2K^2\eta_c^2}{4\sigma_G^2}\right)$, the sequence $\{x_r\}$ generated by FAST satisfies:*

$$\mathbb{E}\|\boldsymbol{x}_R - \boldsymbol{x}^*\|^2 \leq \exp\left(-\mu KR\eta_c\right)\kappa + \frac{(1-q)}{2\mu}K\eta_c + \frac{8}{\mu}K\eta_c\sigma_G^2 + \frac{2(qn+(1-q)m)}{mn\mu}\eta_c\sigma^2,$$

*where $\kappa = \|\boldsymbol{x}_0 - \boldsymbol{x}^*\|^2$ and $\boldsymbol{x}^*$ is the optimal solution.*

**Corollary 3** *For Theorem 2, supposing $\mu > 0$, $m = n$, $\eta_c \leq \frac{1}{20mLK}$ and $R \geq 20mL$, we can obtain*

$$\mathbb{E}\|\boldsymbol{x}_R - \boldsymbol{x}^*\|^2 \leq \widetilde{\mathcal{O}}\left(\exp\left(-\frac{\mu R}{20mL}\right)\right) + \widetilde{\mathcal{O}}\left(\frac{(1-q)}{\mu R}\right) + \widetilde{\mathcal{O}}\left(\frac{1}{\mu R}\sigma_G^2\right) + \widetilde{\mathcal{O}}\left(\frac{1}{\mu mKR}\sigma^2\right),$$

*where $\widetilde{\mathcal{O}}(\cdot)$ subsumes all log-terms and constants. Accordingly, FAST achieves a convergence rate of $\widetilde{\mathcal{O}}(1/R)$.*

**Remark 3** In strongly-convex functions, FAST can achieve a faster convergence rate of $\widetilde{\mathcal{O}}(1/R)$ compared to the non-convex case. It is worth mentioning that this rate can match these rates achieved in FL with ideal client participation (Li et al., 2019). In conjunction with Corollary 1, it is clear that, under appropriate hyper-parameter settings, our FAST can achieve the same convergence rate under ACP as these FL algorithms with ideal client participation.

### 4.3 ADAPTIVE FAST

As shown in Algorithm 1, FAST introduces an extra hyper-parameter, $q$ (or $I$), representing the snapshot probability (or frequency). Obviously, the effective performance of our FAST is evidently contingent on the selection of an appropriate $q$, as indicated by the ablation study on $q$ in Sec. 5. In practice, obtaining prior knowledge to consistently set the optimal $q$ poses a challenge. To address this issue, we propose an adaptive strategy to dynamically update $q$ as shown in Algorithm 2.

---

**Algorithm 2** Adaptive $q$ in FAST

1: Initialize $q_0 = 0$, $\Delta = 0$, $\lambda(default = 1)$.
2: **for** round $r = 0, 1, ..., R - 1$ **do**
3:    Obtain $acc_r$
4:    $\Delta \leftarrow \Delta - acc_r$
5:    $q_{r+1} \leftarrow \min(1, \max(0, q_r + \lambda\Delta))$
6:    $\Delta \leftarrow acc_r$
7: **end for**

---

In more detail, we initiate with $q = 0$ to refrain from enforcing client participation at the beginning of training procedure. Meanwhile, $q$ is adjusted in each round based on the training accuracy difference $\Delta$ between the current and previous rounds. When $\Delta > 0$, indicating a decrease in training accuracy compared to the last round, we increase $q$ by $\lambda\Delta$. This adjustment aims to increase the probability of uniform client participation, improving performance. Conversely, when $\Delta < 0$, signifying an increase in training accuracy in the current round, we decrease $q$ by $\lambda\Delta$. This reduction aims to diminish the probability of uniform client participation, ensuring a more substantial contribution from arbitrary participation in the training process. Line 5 ensures that the frequency $q$ stays within the range of [0,1]. For the selection of $\lambda$, we conduct a series of experiments to assess the performance under different $\lambda$. Our results show that the adaptive FAST is less sensitive to the choice of $\lambda$, and choosing a default $\lambda = 1$ works well under different settings provided in Sec. 5 and Appendix B.2.3.

## 5 EXPERIMENTS

We provide our experiment settings and main results in Sec. 5.1, while leaving other details to Appendix B.2 due to a lack of space.

**Datasets and Models.** We employ Fashion-MNIST (Xiao et al., 2017) and CIFAR-10 datasets (Krizhevsky et al., 2009) for image classification tasks. Additionally, we utilize the Shakespeare dataset (Caldas et al., 2018) for the next character prediction task. For image classification tasks, we train convolutional neural network (CNN) models in our FL system, but the models are different for these two datasets, aiming to adapt to the characteristics of different tasks. Besides, we train the Char-LSTM model for character prediction tasks. Due to space constraints, we direct readers to the Appendix B.1 for comprehensive details regarding datasets and models.

**FL System.** Our FL system comprises 100 clients in total for Fashion-MNIST and CIFAR-10 and 139 clients for Shakespeare. In each round, only $10\%$ clients are chosen to participate in the training process. 1) Data Heterogeneity. The experiments on Fashion-MNIST and CIFAR-10 adhere to balanced and Non-IID datasets. This implies that each client possesses an equal number of data, yet the label distributions differ across clients. To establish this setup, we leverage the FedLab framework (Zeng et al., 2023) for data partitioning. We employ Dirichlet Distribution, as in previous works (Hsu et al., 2019), to generate label-based distributions for each client. By adjusting $\alpha$, we can control the Non-IID degree of data. Generally, a smaller $\alpha$ corresponds to higher data heterogeneity. We provide a data visualization in Figure 1 in Appendix for reference. As for the Shakespeare dataset, its inherent nature is Non-IID. Consequently, we distribute the data of each user to each individual client, ensuring that the number of users in the dataset equals the number of clients in our FL system. 2) Client Participation. We employ four distributions to simulate various participation patterns: uniform,

Beta, Gamma, and Weibull distributions. The uniform distribution serves as ideal client participation. In contrast, the other three distributions act as proxies for ACP. 3) Algorithms. We implement three baselines: FedAvg, FedProx and FedAvgM. Here, we primarily present FedAvg's results, deferring other results to Appendix B.2. When $q = 0$, our FAST becomes the classic FedAvg under various ACP. When $q = 1$, it is the FedAvg under ideal client participation.

Table 3: Experiment results of FAST+FedAvg under different client participation and Non-IID cases.

| Participation | q | Fashion-MNIST | | | | CIFAR-10 | | Shakespeare | |
| | | $\alpha$=0.05 | | $\alpha$=0.1 | | $\alpha$=0.1 | | $\alpha$ =N/A | |
| | | Test Accuracy | Ratio | Test Accuracy | Ratio | Test Accuracy | Ratio | Test Accuracy | Ratio |
|---|---|---|---|---|---|---|---|---|---|
| Uniform (FedAvg) | 1 | 84.10%±2.4 | 0% | 86.85%±1.9 | 0% | 80.18%±0.6 | 0% | 48.86%±0.3 | 0% |
| | Ada.(7) | 80.92%±3.1 | 60.3% | 83.77%±1.8 | 69.4% | 76.83%±1.0 | 67.5% | 48.80%±0.3 | 54.2% |
| | Ada.(def.) | 77.93%±0.7 | 88.5% | 81.08%±1.2 | 95.5% | 68.94%±4.0 | 96.6% | 47.51%±0.6 | 93.9% |
| | 0.5 | 80.74%±2.7 | 49.6% | 82.83%±2.0 | 50.2% | 78.03%±1.3 | 50.7% | 48.63%±0.3 | 49.6% |
| Beta (FAST) | 0.4 | 80.89%±1.3 | 59.6% | 84.07%±2.2 | 60.7% | 78.04%±0.2 | 60.3% | 48.52%±0.3 | 60.1% |
| | 0.3 | 75.88%±4.4 | 69.9% | 81.40%±3.5 | 70.3% | 76.84%±0.6 | 70.1% | 48.31%±0.3 | 70.5% |
| | 0.2 | 76.78%±1.5 | 77.9% | 80.61%±1.3 | 82.6% | 73.90%±1.2 | 80.0% | 47.96%±0.4 | 79.5% |
| | 0.1 | 74.42%±5.3 | 90.9% | 81.15%±0.7 | 88.5% | 72.98%±1.4 | 89.9% | 47.45%±0.6 | 90.2% |
| Beta (FedAvg) | 0 | 74.84%±1.2 | 100% | 79.89%±4.0 | 100% | 68.30%±0.9 | 100% | 46.84%±0.4 | 100% |
| | Ada.(7) | 79.95%±4.9 | 59.3% | 84.42%±2.7 | 74.0% | 76.26%±1.4 | 66.1% | 48.88%±0.3 | 50.8% |
| | Ada.(def.) | 71.48%±4.5 | 91.8% | 82.00%±2.1 | 96.9% | 73.47%±0.5 | 97.3% | 45.37%±0.5 | 92.7% |
| | 0.5 | 77.39%±2.7 | 50.4% | 85.52%±2.3 | 52.0% | 77.76%±0.5 | 49.6% | 48.66%±0.3 | 49.7% |
| Gamma (FAST) | 0.4 | 77.69%±3.5 | 61.0% | 84.23%±2.7 | 59.5% | 78.45%±0.7 | 59.9% | 48.35%±0.4 | 60.7% |
| | 0.3 | 76.87%±2.6 | 68.5% | 85.91%±1.7 | 69.4% | 75.67%±1.1 | 70.7% | 47.69%±0.8 | 69.8% |
| | 0.2 | 72.40%±4.7 | 79.3% | 84.40%±3.1 | 78.2% | 75.70%±0.7 | 81.0% | 47.11%±0.6 | 80.0% |
| | 0.1 | 72.23%±3.2 | 89.7% | 84.55%±2.4 | 91.1% | 74.77%±0.6 | 89.7% | 45.91%±0.7 | 90.3% |
| Gamma (FedAvg) | 0 | 66.65%±4.7 | 100% | 81.81%±1.8 | 100% | 70.90%±0.8 | 100% | 44.46%±1.0 | 100% |
| | Ada.(7) | 77.89%±3.3 | 59.5% | 83.72%±2.7 | 77.3% | 76.37%±1.3 | 66.6% | 48.38%±0.3 | 47.9% |
| | Ada.(def.) | 77.14%±2.7 | 90.4% | 79.83%±3.6 | 97.0% | 72.91%±0.4 | 97.4% | 46.36%±0.8 | 89.0% |
| | 0.5 | 79.10%±4.2 | 50.7% | 85.54%±0.7 | 49.8% | 79.17%±1.0 | 50.2% | 48.55%±0.3 | 50.1% |
| Weibull (FAST) | 0.4 | 77.72%±3.4 | 60.3% | 84.85%±1.1 | 62.3% | 77.70%±0.5 | 59.5% | 48.05%±0.7 | 60.4% |
| | 0.3 | 77.08%±4.2 | 71.3% | 84.48%±2.3 | 69.5% | 75.80%±0.6 | 69.9% | 47.95%±0.4 | 70.4% |
| | 0.2 | 75.66%±4.4 | 80.8% | 81.37%±4.1 | 80.5% | 75.21%±0.6 | 79.1% | 47.27%±0.4 | 80.2% |
| | 0.1 | 75.36%±2.6 | 89.5% | 82.42%±2.5 | 89.4% | 74.14%±1.1 | 89.8% | 46.63%±1.4 | 90.6% |
| Weibull (FedAvg) | 0 | 73.15%±5.1 | 100% | 78.78%±1.6 | 100% | 71.74%±0.7 | 100% | 45.18%±1.8 | 100% |

**Hyper-parameter Settings.** 1) Fashion-MNIST: rounds = 1000, learning rate = 1e-3, training batch size = 128, testing batch size = 1000. 2) CIFAR-10: rounds = 10,000, learning rate = 1e-2, training batch size = 128, testing batch size = 256. At the 5000th round, the learning rate decays by half. 3) Shakespeare: rounds = 5000, learning rate = 0.5, training batch size = 128, testing batch size = 256.

**Note.** For simplicity and clarity, we declare the following notations about experiment results across all tables in this paper: a) Ada.($\lambda$) means adaptive FAST with a fixed $\lambda$. The default $\lambda$ is $\lambda = 1$ denoted by $Ada.(def.)$. b) We define "*Ratio*" as $\frac{\text{Rounds with ACP}}{\text{Total rounds}}$ represents the percentage of arbitrary client participation. $(1 - Ratio)$ represents the percentage of the snapshot enforcement. When $Ratio = 0$, it implies that every round necessitates uniformly random sampling. Conversely, when $Ratio = 1$, only ACP occurs. c) $\alpha$ is the concentration parameter of Dirichlet distribution to control the Non-IID level. d) $A \pm B$: $A$ is the average of the last 5 test accuracy, and $B$ is the standard deviation.

## 5.1 EXPERIMENT RESULTS

In this subsection, we provide four key findings to validate our algorithm and support theoretical analysis: 1) degraded performance for FL algorithms under ACP, 2) improved performance of our FAST algorithm under ACP, 3) compatible framework of our FAST to integrate with other FL algorithms, and 4) ablation study for the hyper-parameters.

**1. FL algorithms' degraded performance under ACP.** In Sec. 4, we illustrate the non-trivial performance degradation of FedAvg under ACP. It is worth noting that this performance degradation is a universal phenomenon extending beyond FedAvg. This is evident in the FedProx results, as shown in Table 4. Results in the table reveal a discernible gap between ideal client participation (uniform distribution) and ACP (other three distributions), with this gap significantly impacted by the level of data heterogeneity. In Appendix, we present more similar findings for other FL algorithms.

**2. Improved performance of our FAST under ACP.** In Table 3, we present a comparison between FedAvg and our FAST across various client participation and Non-IID scenarios, leading to three key findings: 1) Our FAST improves performance by increasing the snapshot frequency ($q$) across all tasks. We conducted experiments with different fixed values of $q$ and observed that our FAST, when configured with $q = 0.5$, nearly matches the test accuracy of FedAvg under ideal client participation.

In other words, we can enforce uniform client participation in only half of the rounds, allowing the remaining rounds to follow ACP. 2) Our adaptive FAST proves effective, showcasing an increased test accuracy with the least snapshots. For instance, in Fashion-MNIST with $\alpha = 0.05$, with the default $\lambda = 1$, our FAST requires only $1 - 91.8\% = 8.2\%$ snapshot enforcement while elevating accuracy from $66.65\%$ to $71.48\%$ in the Gamma distribution. If we take a more aggressive $\lambda = 7$, the accuracy can be improved to $79.95\%$. 3) Our adaptive FAST achieves a great balance between test accuracy and snapshot frequency. Across all cases in Table 3, our default adaptive strategy (Ada.(def.), with $\lambda = 1$) consistently requires less than $10\%$ snapshots while delivering notable improvements.

Table 4: FedProx performance comparison of different client participation in different degrees of Non-IID data.

| Participation \ $\alpha$ | Fashion-MNIST | | | | | CIFAR-10 | | |
|---|---|---|---|---|---|---|---|---|
| | 0.05 | 0.1 | 0.3 | 0.5 | 1.0 | 0.1 | 0.5 | 1.0 |
| Uniform | 83.48%±3.4 | 86.67%±0.5 | 90.36%±1.5 | 91.56%±0.3 | 91.99%±0.6 | 79.53%±0.5 | 80.67%±0.9 | 81.82%±0.5 |
| Beta | 74.74%±0.8 | 77.88%±4.7 | 88.77%±0.4 | 89.44%±0.2 | 89.93%±0.2 | 72.76%±1.0 | 74.76%±0.7 | 76.54%±0.2 |
| Gamma | 75.59%±5.5 | 84.49%±1.9 | 89.76%±1.0 | 89.97%±1.0 | 91.47%±0.2 | 65.94%±1.3 | 79.04%±0.9 | 80.00%±0.3 |
| Weibull | 79.55%±2.0 | 83.66%±2.3 | 89.77%±0.3 | 90.31%±0.7 | 90.97%±0.2 | 72.94%±1.2 | 77.30%±0.3 | 77.63%±0.3 |

Table 5: Performance comparison of different $\lambda$ for adaptive FAST+FedAvg with $\alpha = 0.1$.

| Distribution \ $\lambda$ | 1 | 2 | 3 | 4 | 5 | 6 | 7 | 8 | 9 |
|---|---|---|---|---|---|---|---|---|---|
| Beta (FAST) | 81.08%±1.2 | 80.79%±1.5 | 81.37%±2.3 | 82.40%±1.8 | 81.63%±1.7 | 81.99%±0.7 | 83.77%±1.8 | 81.84%±1.8 | 79.63%±5.2 |
| Ratio | 95.5% | 90.7% | 85.2% | 80.4% | 72.7% | 73.1% | 69.4% | 68.5% | 63.9% |
| Gamma (FAST) | 82.00%±2.1 | 79.63%±5.2 | 82.76%±3.4 | 84.43%±0.7 | 84.47%±1.0 | 86.02%±0.5 | 84.42%±2.7 | 84.55%±1.4 | 83.89%±1.9 |
| Ratio | 96.9% | 90.4% | 86.4% | 82.3% | 77.2% | 77.3% | 74.0% | 71.4% | 70.8% |
| Weibull (FAST) | 79.83%±3.6 | 81.76%±4.6 | 78.88%±2.5 | 82.09%±3.1 | 84.01%±2.0 | 83.29%±2.6 | 83.72%±2.7 | 84.65%±3.7 | 84.84%±1.6 |
| Ratio | 97.0% | 93.0% | 91.2% | 85.8% | 82.3% | 80.3% | 77.3% | 74.8% | 72.7% |

**3. Compatible framework of our FAST to integrate with other FL algorithms.** We highlight that the client participation mechanism in FAST constitutes a general and compatible framework which can seamlessly integrate with other FL algorithms. To demonstrate this, we adopt two additional FL algorithms, FedProx and FedAvgM, utilizing our FAST client participation mechanism, referred to as FAST+. Detailed experimental results are provided in Table 6 and Table 7 in the Appendix due to space limitations. In general, we observe that, under ACP, as modeled by the latter three distributions, the performance of FAST+ significantly surpasses that of FedProx and FedAvgM. These results hold for both fixed $q$ and adaptive $q$. In specific cases, adaptive FAST+ achieves higher test accuracy than FAST+ with a fixed $q$ when their individual proportions of ACP are approximately equal. In other words, adaptive FAST+ can attain higher test accuracy with a higher percentage of ACP (bigger *Ratio* or smaller $q$). These observations align with the results in Table 3 for FedAvg, demonstrating that the client participation mechanism in FAST is general and compatible with other FL algorithms.

**4. Ablation study for the hyper-parameters.** We conducted an extensive series of experiments to perform an ablation study on FL and hyper-parameters of our FAST, including $\alpha$, distributions for modeling client participation, adaptive hyper-parameter $\lambda$, etc. Here, we specifically investigate the impact of $\lambda$ as a key hyper-parameter in our adaptive FAST, leaving all other results in the Appendix B.2.3. In Table 5, we present the test accuracy for Fashion-MNIST with $\lambda$ varying from 1 to 9. Overall, FAST's performance exhibits less sensitivity to the choices of different $\lambda$ values under distinct distributions. As increasing $\lambda$, the snapshot frequency rises, resulting in a decreased ratio. This indicates that the $q$ increases with the increase of $\lambda$. However, we observe that the model performance remains stable. Notably, with our default choice of $\lambda = 1$, our FAST attains good test accuracy with only a small percentage of snapshots. Across these three distributions, when $\lambda = 1$, we require less than $5\%$ of snapshot enforcement, validating the effectiveness of our adaptive FAST.

## 6 CONCLUSION

In this paper, we explored the impact of ACP on FL, characterized by an unknown pattern of client participation. We first empirically showed that FL algorithms are significantly impacted by ACP. Afterward, we proposed a lightweight solution, Federated Average with Snapshot (FAST), to unleash the almost ACP for FL. In strongly-convex and non-convex cases, we proved the convergence rates of FAST, which match the rates with those in ideal client participation. Besides, we introduced an adaptive strategy for dynamically configuring the snapshot frequency, tailored to accommodate diverse FL systems. Extensive numerical results showed that our FAST attains significant improvements under the conditions of ACP and can seamlessly integrate with other classic FL algorithms.

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

## A    PROOF OF THEOREMS

### A.1    LEMMAS

Here we introduce some lemmas that we use in our proof and are also commonly used in other works.

**Lemma 1** *For any $\boldsymbol{x}$ and $\boldsymbol{y}$, L-Lipschitz function F satisfies $F(\boldsymbol{y}) \leq F(\boldsymbol{x}) + \nabla F(\boldsymbol{x})^T(\boldsymbol{y} - \boldsymbol{x}) + \frac{L}{2}\|\boldsymbol{y} - \boldsymbol{x}\|^2$.*

**Lemma 2** *(Karimireddy et al., 2020) For any $\boldsymbol{x}, \boldsymbol{y}, \mathbf{z} \in \boldsymbol{dom}\ g$, any L-smooth and $\mu$-strongly-convex function satisfies $\langle \nabla g(\boldsymbol{x}), \mathbf{z} - \boldsymbol{y} \rangle \geq g(\mathbf{z}) - g(\boldsymbol{y}) + \frac{\mu}{4}\|\boldsymbol{y} - \mathbf{z}\|^2 - L\|\mathbf{z} - \boldsymbol{x}\|^2$.*

### A.2    PROOF OF THEOREM 1

Theorem 1: (Convergence of FAST for Non-convex Functions) *Under the Assumptions 1,2 and 3, supposing that the local step size $\eta_c \leq \min\left\{\frac{1}{8LK}, \frac{nq+m(1-q)}{5mnLK}\right\}$ and the probability $q \geq \frac{(2LK\eta_c-1)G_2+2K^2\sigma_G^2}{G_1+(2LK\eta_c-1)G_2-2LK\eta_cG_3+2K^2\sigma_G^2}$, then the sequence $\{\boldsymbol{x}_r\}$ generated by FAST satisfies:*

$$\frac{1}{R}\sum_{r=1}^{R}\mathbb{E}\|\nabla F(\boldsymbol{x}_r)\|^2 \leq \frac{4\big(F(\boldsymbol{x}_0) - F(\boldsymbol{x}^*)\big)}{KR\eta_c} + \left(\frac{4\big(qn + (1-q)m\big)}{mn}\right)L\eta_c\sigma^2$$

$$+ \Big(120(1-q) + 60q\Big)L^2K^2\eta_c^2\sigma_G^2,$$

where $F(\boldsymbol{x}^*)$ is the minimum value of $F(\cdot)$, and $G_{1-3}$ are defined in the following proof.

**Proof:**

We define that there are totally $R_u = |\mathcal{T}_u| = qR$ rounds under uniform client participation, $R_a = |\mathcal{T}_a| = (1-q)R$ rounds under arbitrary client participation, and total rounds $R = R_u + R_a$.

When the server *uniformly* samples clients at random as a client set $\mathcal{S}_r^u$ and $|\mathcal{S}_r^u| = m$, we begin with Lemma 1 to get

$$\mathbb{E}_r[F(\boldsymbol{x}_{r+1})] \leq F(\boldsymbol{x}_r) + \langle \nabla F(\boldsymbol{x}_r), \mathbb{E}_r[\boldsymbol{x}_{r+1} - \boldsymbol{x}_r]\rangle + \frac{L}{2}\mathbb{E}_r\|\boldsymbol{x}_{r+1} - \boldsymbol{x}_r\|^2$$

$$= F(\boldsymbol{x}_r) + \underbrace{\langle \nabla F(\boldsymbol{x}_r), \mathbb{E}_r[\Delta_r^u]\rangle}_{A_1} + \underbrace{\frac{L}{2}\mathbb{E}_r\|\Delta_r^u\|^2}_{A_2}, \tag{5}$$

where we denote that

$$\Delta_r^u = \boldsymbol{x}_{r+1} - \boldsymbol{x}_r = -\frac{\eta_c}{m}\sum_{i \in \mathcal{S}_r^u}\sum_{k=1}^{K}\nabla F_i(\boldsymbol{x}_{r,k}^i, \xi_{r,k}^i) \text{ and } \mathbb{E}_r[\Delta_r^u] = -\frac{\eta_c}{M}\sum_{i=1}^{M}\sum_{k=1}^{K}\mathbb{E}_r\left[\nabla F_i(\boldsymbol{x}_{r,k}^i)\right].$$

Note that $\mathbb{E}_r[\cdot]$ means the expectation given all randomness generated before the $(r+1)$-th round.

Now, we focus on bounding the term $A_1$. We use the Parallelogram Identity to deal with this cross-term, use assumption 1 to get $A_3$, and use Lemma 3 in (Reddi et al., 2020) to get the inequality (6) under the condition $\eta_c \leq \frac{1}{8LK}$.

$$A_1 = \langle \nabla F(\boldsymbol{x}_r), \mathbb{E}_r[\Delta_r^u]\rangle$$

$$= \langle \nabla F(\boldsymbol{x}_r), -\frac{\eta_c}{M}\sum_{i=1}^{M}\sum_{k=1}^{K}\nabla F_i(\boldsymbol{x}_{r,k}^i)\rangle$$

$$= \frac{\eta_c}{2K}\left(-K^2\|\nabla F(\boldsymbol{x}_r)\|^2 - \left\|\frac{1}{M}\sum_{i=1}^{M}\sum_{k=1}^{K}\nabla F_i(\boldsymbol{x}_{r,k}^i)\right\|^2 + \left\|K\nabla F(\boldsymbol{x}_r) - \frac{1}{M}\sum_{i=1}^{M}\sum_{k=1}^{K}\nabla F_i(\boldsymbol{x}_{r,k}^i)\right\|^2\right)$$

$$= -\frac{K\eta_c}{2}\|\nabla F(\boldsymbol{x}_r)\|^2 - \frac{\eta_c}{2K}\left\|\frac{1}{M}\sum_{i=1}^{M}\sum_{k=1}^{K}\nabla F_i(\boldsymbol{x}_{r,k}^i)\right\|^2 + \frac{\eta_c}{2K}\left\|\frac{1}{M}\sum_{i=1}^{M}\sum_{k=1}^{K}\left(\nabla F_i(\boldsymbol{x}_r) - \nabla F_i(\boldsymbol{x}_{r,k}^i)\right)\right\|^2$$

$$\leq -\frac{K\eta_c}{2}\|\nabla F(\boldsymbol{x}_r)\|^2 - \frac{\eta_c}{2K}\left\|\frac{1}{M}\sum_{i=1}^{M}\sum_{k=1}^{K}\nabla F_i(\boldsymbol{x}_{r,k}^i)\right\|^2 + \frac{\eta_c}{2M}\sum_{i=1}^{M}\sum_{k=1}^{K}\left\|\nabla F_i(\boldsymbol{x}_r) - \nabla F_i(\boldsymbol{x}_{r,k}^i)\right\|^2$$

$$\leq -\frac{K\eta_c}{2}\|\nabla F(\boldsymbol{x}_r)\|^2 - \frac{\eta_c}{2K}\left\|\frac{1}{M}\sum_{i=1}^{M}\sum_{k=1}^{K}\nabla F_i(\boldsymbol{x}_{r,k}^i)\right\|^2 + \frac{L^2\eta_c}{2M}\sum_{i=1}^{M}\sum_{k=1}^{K}\underbrace{\left\|\boldsymbol{x}_r - \boldsymbol{x}_{r,k}^i\right\|^2}_{A_3}$$

$$\leq -\frac{K\eta_c}{2}\|\nabla F(\boldsymbol{x}_r)\|^2 - \frac{\eta_c}{2K}\left\|\frac{1}{M}\sum_{i=1}^{M}\sum_{k=1}^{K}\nabla F_i(\boldsymbol{x}_{r,k}^i)\right\|^2$$

$$+ \frac{L^2\eta_c}{2M}\sum_{i=1}^{M}\sum_{k=1}^{K}\left(5K\eta_c^2(\sigma^2 + 6K\sigma_G^2) + 30K^2\eta_c^2\|\nabla F(x_r)\|^2\right) \tag{6}$$

$$= \left(15L^2K^3\eta_c^3 - \frac{K\eta_c}{2}\right)\|\nabla F(x_r)\|^2 - \frac{\eta_c}{2K}\left\|\frac{1}{M}\sum_{i=1}^{M}\sum_{k=1}^{K}\nabla F_i(\boldsymbol{x}_{r,k}^i)\right\|^2$$

$$+ \frac{5}{2}L^2K^2\eta_c^3\sigma^2 + 15L^2K^3\eta_c^3\sigma_G^2 \tag{7}$$

Next, we use Lemma 4 in (Karimireddy et al., 2020) to bound the $A_2$ term.

$$A_2 = \frac{L}{2}\mathbb{E}_r\|\Delta_r^u\|^2$$

$$= \frac{L}{2}\mathbb{E}_r\left\|-\frac{\eta_c}{m}\sum_{i\in\mathcal{S}_r^u}\sum_{k=1}^{K}\nabla F_i(\boldsymbol{x}_{r,k}^i, \xi_{r,k}^i)\right\|^2$$

$$\leq L\eta_c^2\left\|\frac{1}{m}\sum_{i\in\mathcal{S}_r^u}\sum_{k=1}^{K}\nabla F_i(\boldsymbol{x}_{r,k}^i)\right\|^2 + \frac{LK\eta_c^2}{m}\sigma^2 \tag{8}$$

Plugging (7) and (8) into (5), we obtain

$$\mathbb{E}_r[F(\boldsymbol{x}_{r+1})] \leq F(\boldsymbol{x}_r) + \left(15L^2K^3\eta_c^3 - \frac{K\eta_c}{2}\right)\|\nabla F(\boldsymbol{x}_r)\|^2 - \frac{\eta_c}{2K}\left\|\frac{1}{M}\sum_{i=1}^{M}\sum_{k=1}^{K}\nabla F(\boldsymbol{x}_{r,k}^i)\right\|^2$$

$$+ 15L^2K^3\eta_c^3\sigma_G^2 + \frac{5}{2}L^2K^2\eta_c^3\sigma^2 + L\eta_c^2\left\|\frac{1}{m}\sum_{i\in\mathcal{S}_r^u}\sum_{k=1}^{K}\nabla F_i(\boldsymbol{x}_{r,k}^i)\right\|^2 + \frac{LK\eta_c^2}{m}\sigma^2$$

$$= F(\boldsymbol{x}_r) - \left(\frac{K\eta_c}{2} - 15L^2K^3\eta_c^3\right)\|\nabla F(\boldsymbol{x}_r)\|^2 - \frac{\eta_c}{2K}\left\|\frac{1}{M}\sum_{i=1}^{M}\sum_{k=1}^{K}\nabla F(\boldsymbol{x}_{r,k}^i)\right\|^2$$

$$+ 15L^2K^3\eta_c^3\sigma_G^2 + \left(\frac{5}{2}L^2K^2\eta_c^3 + \frac{LK\eta_c^2}{m}\right)\sigma^2 + L\eta_c^2\left\|\frac{1}{m}\sum_{i\in\mathcal{S}_r^u}\sum_{k=1}^{K}\nabla F_i(\boldsymbol{x}_{r,k}^i)\right\|^2 \tag{9}$$

Rearranging (9), we get

$$K\eta_c\left(\frac{1}{2} - 15L^2K^2\eta_c^2\right)\|\nabla F(\boldsymbol{x}_r)\|^2 \leq F(\boldsymbol{x}_r) - \mathbb{E}_r[F(\boldsymbol{x}_{r+1})] - \frac{\eta_c}{2K}\left\|\frac{1}{M}\sum_{i=1}^{M}\sum_{k=1}^{K}\nabla F(\boldsymbol{x}_{r,k}^i)\right\|^2$$

$$+ 15L^2K^3\eta_c^3\sigma_G^2 + \left(\frac{LK\eta_c^2}{m} + \frac{5}{2}L^2K^2\eta_c^3\right)\sigma^2 + L\eta_c^2\left\|\frac{1}{m}\sum_{i\in\mathcal{S}_r^u}\sum_{k=1}^{K}\nabla F_i(\boldsymbol{x}_{r,k}^i)\right\|^2 \tag{10}$$

To lower bound the constant on the left side of (10), we require the learning rate $\eta_c$ to satisfy that $\eta_c \leq \frac{1}{2\sqrt{15}KL}$. Then, we further simplify (10) to obtain

$$\frac{1}{4}K\eta_c\|\nabla F(\boldsymbol{x}_r)\|^2 \leq F(\boldsymbol{x}_r) - \mathbb{E}_r[F(\boldsymbol{x}_{r+1})] - \frac{\eta_c}{2K}\left\|\frac{1}{M}\sum_{i=1}^{M}\sum_{k=1}^{K}\nabla F(\boldsymbol{x}_{r,k}^i)\right\|^2 + 15L^2K^3\eta_c^3\sigma_G^2 \tag{11}$$

$$+ \left(\frac{LK\eta_c^2}{m} + \frac{5}{2}L^2K^2\eta_c^3\right)\sigma^2 + L\eta_c^2\left\|\frac{1}{m}\sum_{i\in\mathcal{S}_r^u}\sum_{k=1}^{K}\nabla F_i(\boldsymbol{x}_{r,k}^i)\right\|^2 \tag{12}$$

Multiplying $\frac{4}{K\eta_c}$ on the both sides of (12), we have

$$\|\nabla F(\boldsymbol{x}_r)\|^2 \leq \frac{4}{K\eta_c}\left(F(\boldsymbol{x}_r) - \mathbb{E}_r[F(\boldsymbol{x}_{r+1})]\right) - \frac{2}{K^2}\left\|\frac{1}{M}\sum_{i=1}^{M}\sum_{k=1}^{K}\nabla F(\boldsymbol{x}_{r,k}^i)\right\|^2 + 60L^2K^2\eta_c^2\sigma_G^2$$

$$+ \left(\frac{4L\eta_c}{m} + 10L^2K\eta_c^2\right)\sigma^2 + \frac{4L\eta_c}{K}\left\|\frac{1}{m}\sum_{i\in\mathcal{S}_r^u}\sum_{k=1}^{K}\nabla F_i(\boldsymbol{x}_{r,k}^i)\right\|^2 \tag{13}$$

We temporarily keep the result (13) there and continue to deal with the next part of arbitrary client participation in the following.

When the server *arbitrarily* samples clients $\mathcal{S}_r^a$ and $|\mathcal{S}_r^a| = n$, we use Lemma 1 to get

$$\mathbb{E}_r[F(\boldsymbol{x}_{r+1})] \leq F(\boldsymbol{x}_r) + \langle\nabla F(\boldsymbol{x}_r), \mathbb{E}_r[\boldsymbol{x}_{r+1} - \boldsymbol{x}_r]\rangle + \frac{L}{2}\mathbb{E}_r\|\boldsymbol{x}_{r+1} - \boldsymbol{x}_r\|^2$$

$$= F(\boldsymbol{x}_r) + \underbrace{\langle\nabla F(\boldsymbol{x}_r), \mathbb{E}_r[\Delta_r^a]\rangle}_{A_4} + \underbrace{\frac{L}{2}\mathbb{E}_r\|\Delta_r^a\|^2}_{A_5}, \tag{14}$$

where we denote that

$$\Delta_r^a = \boldsymbol{x}_{r+1} - \boldsymbol{x}_r = -\frac{\eta_c}{n}\sum_{i\in\mathcal{S}_r^a}\sum_{k=1}^{K}\nabla F_i(\boldsymbol{x}_{r,k}^i, \xi_{r,k}^i) \text{ and } \mathbb{E}_r[\Delta_r^a] = -\frac{\eta_c}{n}\mathbb{E}_r\sum_{i\in\mathcal{S}_r^a}\sum_{k=1}^{K}\left[\nabla F_i(\boldsymbol{x}_{r,k}^i)\right].$$

We deal with the term $A_4$ by using the Parallelogram Identity.

$$A_4 = \langle\nabla F(\boldsymbol{x}_r), \mathbb{E}_r[\Delta_r^a]\rangle$$

$$= \frac{\eta_c}{2K}\left(-K^2\|\nabla F(\boldsymbol{x}_r)\|^2 - \left\|\frac{1}{n}\sum_{i\in\mathcal{S}_r^a}\sum_{k=1}^{K}\nabla F_i(\boldsymbol{x}_{r,k}^i)\right\|^2 + \left\|K\nabla F(\boldsymbol{x}_r) - \frac{1}{n}\sum_{i\in\mathcal{S}_r^a}\sum_{k=1}^{K}\nabla F_i(\boldsymbol{x}_{r,k}^i)\right\|^2\right)$$

$$= -\frac{K\eta_c}{2}\|\nabla F(\boldsymbol{x}_r)\|^2 - \frac{\eta_c}{2K}\left\|\frac{1}{n}\sum_{i\in\mathcal{S}_r^a}\sum_{k=1}^{K}\nabla F_i(\boldsymbol{x}_{r,k}^i)\right\|^2 + \frac{\eta_c}{2K}\left\|\frac{1}{n}\sum_{i\in\mathcal{S}_r^a}\sum_{k=1}^{K}\left(\nabla F(\boldsymbol{x}_r) - \nabla F_i(\boldsymbol{x}_{r,k}^i)\right)\right\|^2$$

$$\leq -\frac{K\eta_c}{2}\|\nabla F(\boldsymbol{x}_r)\|^2 - \frac{\eta_c}{2K}\left\|\frac{1}{n}\sum_{i\in\mathcal{S}_r^a}\sum_{k=1}^{K}\nabla F_i(\boldsymbol{x}_{r,k}^i)\right\|^2 + \frac{\eta_c}{2n}\sum_{i\in\mathcal{S}_r^a}\sum_{k=1}^{K}\underbrace{\|\nabla F(\boldsymbol{x}_r) - \nabla F_i(\boldsymbol{x}_{r,k}^i)\|^2}_{A_6}$$

$$\tag{15}$$

Then, in the following, we use Jensen's Inequality to bound $A_6$ in the first inequality, utilize L-Lipschitz and data heterogeneity assumptions to get the second inequality, and use Lemma 3 in (Reddi et al., 2020) to get (16) under the condition $\eta_c \leq \frac{1}{8LK}$.

$$A_6 = \|\nabla F(\boldsymbol{x}_r) - \nabla F_i(\boldsymbol{x}_{r,k}^i)\|^2$$

$$= \|\nabla F(\boldsymbol{x}_r) - \nabla F_i(\boldsymbol{x}_r) + \nabla F_i(\boldsymbol{x}_r) - \nabla F_i(\boldsymbol{x}_{r,k}^i)\|^2$$

$$\leq 2\|\nabla F(\boldsymbol{x}_r) - \nabla F_i(\boldsymbol{x}_r)\|^2 + 2\|\nabla F_i(\boldsymbol{x}_r) - \nabla F_i(\boldsymbol{x}_{r,k}^i)\|^2$$

$$\leq 2\sigma_G^2 + 2L^2\|\boldsymbol{x}_r - \boldsymbol{x}_{r,k}^i\|^2$$

$$\leq 2\sigma_G^2 + 2L^2\Big(5K\eta_c^2(\sigma^2 + 6K\sigma_G^2) + 30K^2\eta_c^2\|\nabla F(x_r)\|^2\Big) \tag{16}$$

$$= 60L^2K^2\eta_c^2\|\nabla F(\boldsymbol{x}_r)\|^2 + (2 + 60L^2K^2\eta_c^2)\sigma_G^2 + 10L^2K\eta_c^2\sigma^2 \tag{17}$$

Plugging (17) into (15), we establish

$$A_4 \leq -\frac{K\eta_c}{2}\|\nabla F(\boldsymbol{x}_r)\|^2 - \frac{\eta_c}{2K}\left\|\frac{1}{n}\sum_{i\in\mathcal{S}_r^a}\sum_{k=1}^K \nabla F_i(\boldsymbol{x}_{r,k}^i)\right\|^2$$

$$+ \frac{\eta_c}{2n}\sum_{i\in\mathcal{S}_r^a}\sum_{k=1}^K \Big(60L^2K^2\eta_c^2\|\nabla F(\boldsymbol{x}_r)\|^2 + (2 + 60L^2K^2\eta_c^2)\sigma_G^2 + 10L^2K\eta_c^2\sigma^2\Big)$$

$$= -\frac{K\eta_c}{2}\|\nabla F(\boldsymbol{x}_r)\|^2 - \frac{\eta_c}{2K}\left\|\frac{1}{n}\sum_{i\in\mathcal{S}_r^a}\sum_{k=1}^K \nabla F_i(\boldsymbol{x}_{r,k}^i)\right\|^2 + 30L^2K^3\eta_c^3\|\nabla F(\boldsymbol{x}_r)\|^2$$

$$+ (K\eta_c + 30L^2K^3\eta_c^3)\sigma_G^2 + 5L^2K^2\eta_c^3\sigma^2$$

$$= \Big(30L^2K^3\eta_c^3 - \frac{K\eta_c}{2}\Big)\|\nabla F(\boldsymbol{x}_r)\|^2 - \frac{\eta_c}{2K}\left\|\frac{1}{n}\sum_{i\in\mathcal{S}_r^a}\sum_{k=1}^K \nabla F_i(\boldsymbol{x}_{r,k}^i)\right\|^2$$

$$+ (K\eta_c + 30L^2K^3\eta_c^3)\sigma_G^2 + 5L^2K^2\eta_c^3\sigma^2 \tag{18}$$

Next, we apply Lemma 4 in (Karimireddy et al., 2020) to bound the $A_5$ term.

$$A_5 = \frac{L}{2}\mathbb{E}_r\|\Delta_r\|^2$$

$$= \frac{L\eta_c^2}{2}\mathbb{E}_r\left\|\frac{1}{n}\sum_{i\in\mathcal{S}_r^a}\sum_{k=1}^K \nabla F_i(\boldsymbol{x}_{r,k}^i, \xi_{r,k}^i)\right\|^2$$

$$\leq L\eta_c^2\left\|\frac{1}{n}\sum_{i\in\mathcal{S}_r^a}\sum_{k=1}^K \nabla F_i(\boldsymbol{x}_{r,k}^i)\right\|^2 + \frac{LK\eta_c^2}{n}\sigma^2 \tag{19}$$

Plugging (18) and (19) into (14), we obtain

$$\mathbb{E}_r[F(\boldsymbol{x}_{r+1})] \leq F(\boldsymbol{x}_r) + \Big(30L^2K^3\eta_c^3 - \frac{K\eta_c}{2}\Big)\|\nabla F(\boldsymbol{x}_r)\|^2 - \frac{\eta_c}{2K}\left\|\frac{1}{n}\sum_{i\in\mathcal{S}_r^a}\sum_{k=1}^K \nabla F_i(\boldsymbol{x}_{r,k}^i)\right\|^2$$

$$+ (K\eta_c + 30L^2K^3\eta_c^3)\sigma_G^2 + 5L^2K^2\eta_c^3\sigma^2 + L\eta_c^2\left\|\frac{1}{n}\sum_{i\in\mathcal{S}_r^a}\sum_{k=1}^K \nabla F_i(\boldsymbol{x}_{r,k}^i)\right\|^2 + \frac{LK\eta_c^2}{n}\sigma^2$$

$$= F(\boldsymbol{x}_r) - \Big(\frac{K\eta_c}{2} - 30L^2K^3\eta_c^3\Big)\|\nabla F(\boldsymbol{x}_r)\|^2 - \Big(\frac{\eta_c}{2K} - L\eta_c^2\Big)\left\|\frac{1}{n}\sum_{i\in\mathcal{S}_r^a}\sum_{k=1}^K \nabla F_i(\boldsymbol{x}_{r,k}^i)\right\|^2$$

$$+ (K\eta_c + 30L^2K^3\eta_c^3)\sigma_G^2 + \Big(\frac{LK\eta_c^2}{n} + 5L^2K^2\eta_c^3\Big)\sigma^2 \tag{20}$$

Rearranging (20), we establish

$$\Big(\frac{K\eta_c}{2} - 30L^2K^3\eta_c^3\Big)\|\nabla F(\boldsymbol{x}_r)\|^2 \leq F(\boldsymbol{x}_r) - \mathbb{E}_r[F(\boldsymbol{x}_{r+1})] - \Big(\frac{\eta_c}{2K} - L\eta_c^2\Big)\left\|\frac{1}{n}\sum_{i\in\mathcal{S}_r^a}\sum_{k=1}^K \nabla F_i(\boldsymbol{x}_{r,k}^i)\right\|^2$$

$$+ (K\eta_c + 30L^2K^3\eta_c^3)\sigma_G^2 + \left(\frac{LK\eta_c^2}{n} + 5L^2K^2\eta_c^3\right)\sigma^2 \tag{21}$$

To lower bound the constant on the left side of (21), we require the learning rate $\eta_c$ to satisfy that $\eta_c \leq \frac{1}{2\sqrt{30}KL}$.

Then, we arrive at

$$\frac{1}{4}K\eta_c\|\nabla F(\boldsymbol{x}_r)\|^2 \leq F(\boldsymbol{x}_r) - \mathbb{E}_r[F(\boldsymbol{x}_{r+1})] - \left(\frac{\eta_c}{2K} - L\eta_c^2\right)\left\|\frac{1}{n}\sum_{i\in\mathcal{S}_r^a}\sum_{k=1}^K \nabla F_i(\boldsymbol{x}_{r,k}^i)\right\|^2$$

$$+ (K\eta_c + 30L^2K^3\eta_c^3)\sigma_G^2 + \left(\frac{LK\eta_c^2}{n} + 5L^2K^2\eta_c^3\right)\sigma^2 \tag{22}$$

Multiplying $\frac{4}{K\eta_c}$ on the both sides of (22), we obtain

$$\|\nabla F(\boldsymbol{x}_r)\|^2 \leq \frac{4}{K\eta_c}\Big(F(\boldsymbol{x}_r) - \mathbb{E}_r[F(\boldsymbol{x}_{r+1})]\Big) - \left(\frac{2}{K^2} - \frac{4L\eta_c}{K}\right)\left\|\frac{1}{n}\sum_{i\in\mathcal{S}_r^a}\sum_{k=1}^K \nabla F_i(\boldsymbol{x}_{r,k}^i)\right\|^2$$

$$+ (4 + 120L^2K^2\eta_c^2)\sigma_G^2 + \left(\frac{4L\eta_c}{n} + 20L^2K\eta_c^2\right)\sigma^2 \tag{23}$$

Note that there are total $R$ ($R = R_u + R_a$) rounds, including $R_u$ rounds ($\mathcal{T}_u$ as the round indices) under uniform client participation and $R_a$ rounds ($\mathcal{T}_a$ as the round indices) under arbitrary client participation. Then, by (13) from uniform participation and (23) from arbitrary participation, executing our algorithm for $R$ rounds, we can establish

$$\frac{1}{R}\sum_{r=1}^R \|\nabla F(\boldsymbol{x}_r)\|^2 \leq \frac{4}{K\eta_c}\frac{1}{R}\sum_{r\in\mathcal{S}_r^u}\Big(F(\boldsymbol{x}_r) - \mathbb{E}_r[F(\boldsymbol{x}_{r+1})]\Big) - \frac{2}{K^2}\frac{1}{R}\sum_{r\in\mathcal{S}_r^u}\left\|\frac{1}{M}\sum_{i=1}^M\sum_{k=1}^K \nabla F(\boldsymbol{x}_{r,k}^i)\right\|^2$$

$$+ \frac{1}{R}\sum_{r\in\mathcal{S}_r^u} 60L^2K^2\eta_c^2\sigma_G^2 + \frac{1}{R}\sum_{r\in\mathcal{S}_r^u}\left(\frac{4L\eta_c}{m} + 10L^2K\eta_c^2\right)\sigma^2$$

$$+ \frac{4L\eta_c}{K}\frac{1}{R}\sum_{r\in\mathcal{S}_r^u}\left\|\frac{1}{m}\sum_{i\in\mathcal{S}_r^u}\sum_{k=1}^K \nabla F_i(\boldsymbol{x}_{r,k}^i)\right\|^2 + \frac{4}{K\eta_c}\frac{1}{R}\sum_{r\in\mathcal{S}_r^a}\Big(F(\boldsymbol{x}_r) - \mathbb{E}_r[F(\boldsymbol{x}_{r+1})]\Big)$$

$$- \left(\frac{2}{K^2} - \frac{4L\eta_c}{K}\right)\frac{1}{R}\sum_{r\in\mathcal{S}_r^a}\left\|\frac{1}{n}\sum_{i\in\mathcal{S}_r^a}\sum_{k=1}^K \nabla F_i(\boldsymbol{x}_{r,k}^i)\right\|^2$$

$$+ \frac{1}{R}\sum_{r\in\mathcal{S}_r^a}(4 + 120L^2K^2\eta_c^2)\sigma_G^2 + \frac{1}{R}\sum_{r\in\mathcal{S}_r^a}\left(\frac{4L\eta_c}{n} + 20L^2K\eta_c^2\right)\sigma^2$$

$$\leq \frac{4\big(F(\boldsymbol{x}_0) - F(\boldsymbol{x}^*)\big)}{KR\eta_c} + \left(\frac{4\big(qn + (1-q)m\big)}{mn}\right)L\eta_c\sigma^2$$

$$+ \Big(10q + 20(1-q)\Big)L^2K\eta_c^2\sigma^2 + \Big(120(1-q) + 60q\Big)L^2K^2\eta_c^2\sigma_G^2$$

$$\leq \frac{4\big(F(\boldsymbol{x}_0) - F(\boldsymbol{x}^*)\big)}{KR\eta_c} + \left(\frac{4\big(qn + (1-q)m\big)}{mn}\right)L\eta_c\sigma^2$$

$$+ \Big(120(1-q) + 60q\Big)L^2K^2\eta_c^2\sigma_G^2, \tag{24}$$

where the inequality (24) holds with following conditions: learning rate $\eta_c \leq \frac{nq + m(1-q)}{5mnLK}$ and $q \geq \frac{(2LK\eta_c - 1)G_2 + 2K^2\sigma_G^2}{G_1 + (2LK\eta_c - 1)G_2 - 2LK\eta_cG_3 + 2K^2\sigma_G^2}$. For the domain of $q$, we introduce

$$G_1 \quad := \quad \max_{r \in \mathcal{T}_u} \left\| \frac{1}{M} \sum_{i=1}^{M} \sum_{k=1}^{K} \nabla F_i(\boldsymbol{x}_{r,k}^i) \right\|^2, \quad G_2 \quad := \quad \max_{r \in \mathcal{T}_a} \left\| \frac{1}{n} \sum_{i \in \mathcal{S}_r^a} \sum_{k=1}^{K} \nabla F_i(\boldsymbol{x}_{r,k}^i) \right\|^2 \text{ and } G_3 \quad :=$$

$$\max_{r \in \mathcal{T}_u} \left\| \frac{1}{m} \sum_{i \in \mathcal{S}_r^u} \sum_{k=1}^{K} \nabla F_i(\boldsymbol{x}_{r,k}^i) \right\|^2 \text{ to light the notation.}$$

Here, for clarification, we further explain how we get the inequality (24) and the domain of $q$ above. The key idea is to set $-\frac{2}{K^2} \frac{1}{R} G_1 + \frac{4L\eta_C}{K} \frac{1}{R} G_3 - (\frac{2}{K^2} - \frac{4L\eta_c}{K} \frac{1}{R}) G_2 + \frac{1}{R} \sum_{r \in \mathcal{S}_a} 4\sigma_G^2 \leq 0$, and then we can drop this entire negative term and follow the condition $q \geq \frac{(2LK\eta_c - 1)G_2 + 2K^2\sigma_G^2}{G_1 + (2LK\eta_c - 1)G_2 - 2LK\eta_c G_3 + 2K^2\sigma_G^2}$.

To further get the convergence rate, we set learning rate to satisfy $\eta_c = \mathcal{O}\left( \frac{\sqrt{mn}}{\sqrt{RK(qn + (1-q)m)}} \right)$, and then we can obtain the convergence bound in the non-convex case as follows.

$$\frac{1}{R} \sum_{r=1}^{R} \mathbb{E} \left\| \nabla F(\boldsymbol{x}_r) \right\|^2 \leq \mathcal{O}\left( \sqrt{\frac{nq + m(1-q)}{nmKR}} \right) + \mathcal{O}\left( \sqrt{\frac{nq + m(1-q)}{mnRK}} \sigma^2 \right)$$
$$+ \mathcal{O}\left( \frac{mnK}{(nq + (1-q)m)R} \sigma_G^2 \right)$$

$\blacksquare$

### A.3 PROOF OF THEOREM 2

**Theorem 2: (Convergence of FAST for Strongly Convex Functions)** *Under assumptions 1,2,3 and 4, if the learning rate and probability of snapshots satisfy* $0 < \eta_c \leq \min\left\{ \frac{1}{20mLK}, \frac{1}{20nLK} \right\}$ *and* $q \geq 1 - \left( \frac{\mu K \eta_c - 16L^2K^2\eta_c^2}{4} \right)$, *then the sequence* $\{x_r\}$ *generated by FAST satisfies:*

$$\mathbb{E} \left\| \boldsymbol{x}_R - \boldsymbol{x}^* \right\|^2 \leq \exp\left( -\mu KR\eta_c \right) \kappa + \frac{(1-q)}{2\mu} K\eta_c \sigma_G^2 + \frac{8}{\mu} K\eta_c \sigma_G^2 + \frac{2(qn + (1-q)m)}{mn\mu} \eta_c \sigma^2, \tag{25}$$

*where* $\kappa := \left\| \boldsymbol{x}_0 - \boldsymbol{x}^* \right\|^2$ *and* $x^*$ *is the optimal solution.*

**Proof:**

In this proof, we still use the same definitions as the Proof A.2. There are totally $R_u = |\mathcal{T}_u| = qR$ rounds under uniform client participation, $R_a = |\mathcal{T}_a| = (1-q)R$ rounds under arbitrary client participation, and total rounds $R = R_u + R_a$.

First, we deal with the uniform client participation part. When the server *uniformly* samples clients at random in $r$-th round as a client set $\mathcal{S}_r^u$ and $|\mathcal{S}_r^u| = m$, we have

$$\mathbb{E}_r^u \left\| \boldsymbol{x}_{r+1} - \boldsymbol{x}^* \right\|^2 = \mathbb{E}_r \left\| \boldsymbol{x}_r + \Delta_r^u - \boldsymbol{x}^* \right\|^2$$
$$= \mathbb{E}_r \left\| \boldsymbol{x}_r - \boldsymbol{x}^* \right\|^2 + 2\mathbb{E}_r \langle \Delta_r^u, \boldsymbol{x}_r - \boldsymbol{x}^* \rangle + \mathbb{E}_r \left\| \Delta_r^u \right\|^2$$
$$= \left\| \boldsymbol{x}_r - \boldsymbol{x}^* \right\|^2 - 2\eta_c \underbrace{\langle \frac{1}{M} \sum_{i=1}^{M} \sum_{k=1}^{K} \nabla F_i(\boldsymbol{x}_{r,k}^i), \boldsymbol{x}_r - \boldsymbol{x}^* \rangle}_{A_7}$$
$$+ \eta_c^2 \mathbb{E}_r \underbrace{\left\| \frac{1}{m} \sum_{i \in \mathcal{S}_r^u} \sum_{k=1}^{K} \nabla F_i(\boldsymbol{x}_{r,k}^i, \xi_{r,k}^i) \right\|^2}_{A_8}, \tag{26}$$

where we denote that

$$\Delta_r^u = \boldsymbol{x}_{r+1} - \boldsymbol{x}_r = -\frac{\eta_c}{m} \sum_{i \in \mathcal{S}_r^u} \sum_{k=1}^{K} \nabla F_i(\boldsymbol{x}_{r,k}^i, \xi_{r,k}^i) \text{ and } \mathbb{E}_r[\Delta_r^u] = -\frac{\eta_c}{M} \sum_{i=1}^{M} \sum_{k=1}^{K} \mathbb{E}_r \left[ \nabla F_i(\boldsymbol{x}_{r,k}^i) \right].$$

We use lemma 2 to deal with $A_7$ and use the Lemma 3 in Reddi et al. (2020) to get the inequality (27), following the condition $\eta_c \leq \frac{1}{8LK}$

$$
\begin{aligned}
A_7 =& \frac{1}{M} \sum_{i=1}^{M} \sum_{k=1}^{K} \langle \nabla F_i(\boldsymbol{x}_{r,k}^i), \boldsymbol{x}_r - \boldsymbol{x}^* \rangle \\
\geq& \frac{1}{M} \sum_{i=1}^{M} \sum_{k=1}^{K} \left[ F_i(\boldsymbol{x}_r) - F_i(\boldsymbol{x}^*) + \frac{\mu}{4}\|\boldsymbol{x}^* - \boldsymbol{x}_r\|^2 - L\|\boldsymbol{x}_r - \boldsymbol{x}_{r,k}^i\|^2 \right] \\
\geq& \frac{1}{M} \sum_{i=1}^{M} \sum_{k=1}^{K} \left[ \langle \nabla F_i(\boldsymbol{x}^*), \boldsymbol{x}_r - \boldsymbol{x}^* \rangle + \frac{\mu}{2}\|\boldsymbol{x}_r - \boldsymbol{x}^*\|^2 + \frac{\mu}{4}\|\boldsymbol{x}^* - \boldsymbol{x}_r\|^2 - L\|\boldsymbol{x}_r - \boldsymbol{x}_{r,k}^i\|^2 \right] \\
\geq& \frac{3\mu K}{4}\|\boldsymbol{x}_r - \boldsymbol{x}^*\|^2 - LK\left(5K\eta_c^2(\sigma^2 + 6K\sigma_G^2) + 30K^2\eta_c^2\|\nabla F(\boldsymbol{x}_r)\|^2\right) && (27) \\
\geq& \frac{3\mu K}{4}\|\boldsymbol{x}_r - \boldsymbol{x}^*\|^2 - LK\left(5K\eta_c^2(\sigma^2 + 6K\sigma_G^2) + 30L^2K^2\eta_c^2\|\boldsymbol{x}_r - \boldsymbol{x}^*\|^2\right) \\
\geq& \left(\frac{3\mu K}{4} - 30L^3K^3\eta_c^2\right)\|\boldsymbol{x}_r - \boldsymbol{x}^*\|^2 - 5LK^2\eta_c^2\sigma^2 - 30LK^3\eta_c^2\sigma_G^2 && (28)
\end{aligned}
$$

Then, we bound the term $A_8$ as follows. We use Jensen's inequality to get (a), use L-smoothness property to get (b), use Lemma 3 in Reddi et al. (2020) and the data heterogeneity assumption to get (c), use L-smoothness property to get (d).

$$
\begin{aligned}
A_8 =& \eta_c^2 \mathbb{E}_r \left\| \frac{1}{m} \sum_{i \in \mathcal{S}_r^u} \sum_{k=1}^{K} \nabla F_i(\boldsymbol{x}_{r,k}^i, \xi_{r,k}^i) \right\|^2 \\
\leq& \eta_c^2 \mathbb{E}_r \left\| \frac{1}{m} \sum_{i \in \mathcal{S}_r^u} \sum_{k=1}^{K} \nabla F_i(\boldsymbol{x}_{r,k}^i) \right\|^2 + \frac{K\eta_c^2}{m}\sigma^2 \\
=& \eta_c^2 \mathbb{E}_r \left\| \frac{1}{m} \sum_{i \in \mathcal{S}_r^u} \sum_{k=1}^{K} \left[ \nabla F_i(\boldsymbol{x}_{r,k}^i) - \nabla F_i(\boldsymbol{x}_r) + \nabla F_i(\boldsymbol{x}_r) \right] \right\|^2 + \frac{K\eta_c^2}{m}\sigma^2 \\
\overset{(a)}{\leq}& \frac{2K\eta_c^2}{m} \mathbb{E}_r \sum_{i \in \mathcal{S}_r^u} \sum_{k=1}^{K} \left\| \nabla F_i(\boldsymbol{x}_{r,k}^i) - \nabla F_i(\boldsymbol{x}_r) \right\|^2 + 2\eta_c^2 \mathbb{E}_r \left\| \frac{1}{m} \sum_{i \in \mathcal{S}_r^u} \sum_{k=1}^{K} \nabla F_i(\boldsymbol{x}_r) \right\|^2 + \frac{K\eta_c^2}{m}\sigma^2 \\
\overset{(b)}{\leq}& \frac{2L^2K\eta_c^2}{m} \mathbb{E}_r \sum_{i \in \mathcal{S}_r^u} \sum_{k=1}^{K} \left\| \boldsymbol{x}_r - \boldsymbol{x}_{r,k}^i \right\|^2 + 2\eta_c^2 \mathbb{E}_r \left\| \frac{1}{m} \sum_{i \in \mathcal{S}_r^u} \sum_{k=1}^{K} \left[ \nabla F_i(\boldsymbol{x}_r) - \nabla F_i(\boldsymbol{x}^*) + \nabla F_i(\boldsymbol{x}^*) \right] \right\|^2 \\
&+ \frac{K\eta_c^2}{m}\sigma^2 \\
\overset{(c)}{\leq}& 2L^2K^2\eta_c^2 \left( 5K\eta_c^2(\sigma^2 + 6K\sigma_G^2) + 30K^2\eta_c^2\|\nabla F(\boldsymbol{x}_r)\|^2 \right) \\
&+ 4K^2\eta_c^2 \mathbb{E}_r \left\| \frac{1}{m} \sum_{i \in \mathcal{S}_r^u} \left[ \nabla F_i(\boldsymbol{x}_r) - \nabla F_i(\boldsymbol{x}^*) \right] \right\|^2 + 4\eta_c^2 \mathbb{E}_r \left\| \frac{1}{m} \sum_{i \in \mathcal{S}_r^u} \sum_{k=1}^{K} \nabla F_i(\boldsymbol{x}^*) \right\|^2 + \frac{K\eta_c^2}{m}\sigma^2 \\
\leq& 2L^2K^2\eta_c^2 \left( 30K^2\eta_c^2\|\nabla F(\boldsymbol{x}_r)\|^2 + 30K^2\eta_c^2\sigma_G^2 + 5K\eta_c^2\sigma^2 \right) \\
&+ \frac{4K^2\eta_c^2}{m} \mathbb{E}_r \sum_{i \in \mathcal{S}_r^u} \|\nabla F_i(\boldsymbol{x}_r) - \nabla F_i(\boldsymbol{x}^*)\|^2 + 4K^2\eta_c^2\sigma_G^2 + \frac{K\eta_c^2}{m}\sigma^2 \\
\overset{(d)}{\leq}& 2L^2K^2\eta_c^2 \left( 30L^2K^2\eta_c^2\|\boldsymbol{x}_r - \boldsymbol{x}^*\|^2 + 30K^2\eta_c^2\sigma_G^2 + 5K\eta_c^2\sigma^2 \right) + 4L^2K^2\eta_c^2 \|\boldsymbol{x}_r - \boldsymbol{x}^*\|^2 \\
&+ 4K^2\eta_c^2\sigma_G^2 + \frac{K\eta_c^2}{m}\sigma^2
\end{aligned}
$$

$$
\begin{aligned}
=&(4L^2K^2\eta_c^2 + 60L^4K^4\eta_c^4)\|\boldsymbol{x}_r - \boldsymbol{x}^*\|^2 + \left(4K^2\eta_c^2 + 60L^2K^4\eta_c^4\right)\sigma_G^2 \\
&+ \left(\frac{K\eta_c^2}{m} + 10L^2K^3\eta_c^4\right)\sigma^2
\end{aligned}
\tag{29}
$$

Plugging (28) and (29) back to (26) and absorbing higher-order terms into lower-order terms, we obtain

$$
\begin{aligned}
\mathbb{E}_r^u\|\boldsymbol{x}_{r+1} - \boldsymbol{x}^*\|^2 \leq& \|\boldsymbol{x}_r - \boldsymbol{x}^*\|^2 + \left(\frac{K\eta_c^2}{m} + 10L^2K^3\eta_c^4\right)\sigma^2 \\
& - 2\eta_c\left(\left(\frac{3\mu K}{4} - 30L^3K^3\eta_c^2\right)\|\boldsymbol{x}^* - \boldsymbol{x}_r\|^2 - 5LK^2\eta_c^2\sigma^2 - 30LK^3\eta_c^2\sigma_G^2\right) \\
& + (4L^2K^2\eta_c^2 + 60L^4K^4\eta_c^4)\|\boldsymbol{x}_r - \boldsymbol{x}^*\|^2 + (4K^2\eta_c^2 + 60L^2K^4\eta_c^4)\sigma_G^2 \\
\leq& \left(1 - \frac{3\mu K\eta_c}{2} + 8L^2K^2\eta_c^2\right)\|\boldsymbol{x}_r - \boldsymbol{x}^*\|^2 + 8K^2\eta_c^2\sigma_G^2 + \frac{2K\eta_c^2}{m}\sigma^2,
\end{aligned}
\tag{30}
$$

where the last inequality above follows the condition $\eta_c \leq \frac{1}{20mLK}$.

Now, we deal with the part of arbitrary participation. When the server *arbitrarily* samples clients $\mathcal{S}_r^a$ and $|\mathcal{S}_r^a| = n$, we have

$$
\begin{aligned}
\mathbb{E}_r^a\|\boldsymbol{x}_{r+1} - \boldsymbol{x}^*\|^2 =& \mathbb{E}_r\|\boldsymbol{x}_r + \Delta_r^a - \boldsymbol{x}^*\|^2 \\
=& \mathbb{E}_r\|\boldsymbol{x}_r - \boldsymbol{x}^*\|^2 + 2\mathbb{E}_r\langle\Delta_r^a, \boldsymbol{x}_r - \boldsymbol{x}^*\rangle + \mathbb{E}_r\|\Delta_r^a\|^2 \\
=& \|\boldsymbol{x}_r - \boldsymbol{x}^*\|^2 - \underbrace{2\eta_c\mathbb{E}_r\langle\frac{1}{n}\sum_{i\in\mathcal{S}_r^a}\sum_{k=1}^K \nabla F_i(\boldsymbol{x}_{r,k}^i), \boldsymbol{x}_r - \boldsymbol{x}^*\rangle}_{A_9} \\
& + \underbrace{\eta_c^2\mathbb{E}_r\left\|\frac{1}{n}\sum_{i\in\mathcal{S}_r^a}\sum_{k=1}^K \nabla F_i(\boldsymbol{x}_{r,k}^i, \xi_{r,k}^i)\right\|^2}_{A_{10}}
\end{aligned}
\tag{31}
$$

We deal with the term $A_9$ as follows. We use Lemma 2 to obtain (32), use the Young's Inequality to get (33) ($\epsilon > 0$), and use Lemma 3 in Reddi et al. (2020) to obtain (34).

$$
\begin{aligned}
A_9 =& 2\eta_c\mathbb{E}_r\langle\frac{1}{n}\sum_{i\in\mathcal{S}_r^a}\sum_{k=1}^K \nabla F_i(\boldsymbol{x}_{r,k}^i), \boldsymbol{x}_r - \boldsymbol{x}^*\rangle \\
\geq& \frac{2\eta_c}{n}\mathbb{E}_r\sum_{i\in\mathcal{S}_r^a}\sum_{k=1}^K\left[F_i(\boldsymbol{x}_r) - F_i(\boldsymbol{x}^*) + \frac{\mu}{4}\|\boldsymbol{x}_r - \boldsymbol{x}^*\|^2 - L\|\boldsymbol{x}_r - \boldsymbol{x}_{r,k}^i\|^2\right] \\
\geq& \frac{2\eta_c}{n}\mathbb{E}_r\sum_{i\in\mathcal{S}_r^a}\sum_{k=1}^K\left[\langle\nabla F_i(\boldsymbol{x}^*), \boldsymbol{x}_r - \boldsymbol{x}^*\rangle + \frac{\mu}{2}\|\boldsymbol{x}_r - \boldsymbol{x}^*\|^2 + \frac{\mu}{4}\|\boldsymbol{x}_r - \boldsymbol{x}^*\|^2 - L\|\boldsymbol{x}_r - \boldsymbol{x}_{r,k}^i\|^2\right] \\
\geq& \frac{2\eta_c}{n}\mathbb{E}_r\sum_{i\in\mathcal{S}_r^a}\sum_{k=1}^K\left[\langle\nabla F_i(\boldsymbol{x}^*) - \nabla F(\boldsymbol{x}^*), \boldsymbol{x}_r - \boldsymbol{x}^*\rangle + \frac{3\mu}{4}\|\boldsymbol{x}_r - \boldsymbol{x}^*\|^2 - L\|\boldsymbol{x}_r - \boldsymbol{x}_{r,k}^i\|^2\right] \\
\geq& \frac{2K\eta_c}{n}\mathbb{E}_r\sum_{i\in\mathcal{S}_r^a}\langle\nabla F_i(\boldsymbol{x}^*) - \nabla F(\boldsymbol{x}^*), \boldsymbol{x}_r - \boldsymbol{x}^*\rangle + \frac{3\mu K\eta_c}{2}\|\boldsymbol{x}_r - \boldsymbol{x}^*\|^2 \\
& - \frac{2L\eta_c}{n}\mathbb{E}_r\sum_{i\in\mathcal{S}_r^a}\sum_{k=1}^K\|\boldsymbol{x}_r - \boldsymbol{x}_{r,k}^i\|^2 \\
\geq& \frac{2K\eta_c}{n}\mathbb{E}_r\sum_{i\in\mathcal{S}_r^a}\left(-\frac{1}{4\epsilon}\|\nabla F(\boldsymbol{x}^*) - \nabla F_i(\boldsymbol{x}^*)\|^2 - \epsilon\|\boldsymbol{x}_r - \boldsymbol{x}^*\|^2\right) + \frac{3\mu K\eta_c}{2}\|\boldsymbol{x}_r - \boldsymbol{x}^*\|^2
\end{aligned}
$$

$$- \frac{2L\eta_c}{n} \mathbb{E}_r \sum_{i \in \mathcal{S}_r^a} \sum_{k=1}^{K} \|\boldsymbol{x}_r - \boldsymbol{x}_{r,k}^i\|^2 \tag{33}$$

$$\geq - \frac{K\eta_c}{2\epsilon}\sigma_G^2 - 2K\eta_c\epsilon\|\boldsymbol{x}_r - \boldsymbol{x}^*\|^2 + \frac{3\mu K\eta_c}{2}\|\boldsymbol{x}_r - \boldsymbol{x}^*\|^2$$
$$- 2LK\eta_c \left(5K\eta_c^2(\sigma^2 + 6K\sigma_G^2) + 30K^2\eta_c^2\|\nabla F(\boldsymbol{x}_r)\|^2\right) \tag{34}$$

$$\geq - \frac{K\eta_c}{2\epsilon}\sigma_G^2 - 2K\eta_c\epsilon\|\boldsymbol{x}_r - \boldsymbol{x}^*\|^2 + \frac{3\mu K\eta_c}{2}\|\boldsymbol{x}_r - \boldsymbol{x}^*\|^2$$
$$- 2LK\eta_c \left(5K\eta_c^2(\sigma^2 + 6K\sigma_G^2) + 30L^2K^2\eta_c^2\|\boldsymbol{x}_r - \boldsymbol{x}^*\|^2\right)$$

$$\geq - \left(\frac{K\eta_c}{2\epsilon} + 60LK^3\eta_c^3\right)\sigma_G^2 - \left(2K\eta_c\epsilon + 60L^3K^3\eta_c^3 - \frac{3\mu K\eta_c}{2}\right)\|\boldsymbol{x}_r - \boldsymbol{x}^*\|^2 - 10LK^2\eta_c^3\sigma^2 \tag{35}$$

Then, we bound $A_{10}$ as follows. We use Lemma 3 in Reddi et al. (2020) to get (36).

$$A_{10} = \eta_c^2 \mathbb{E}_r \left\| \frac{1}{n} \sum_{i \in \mathcal{S}_r^a} \sum_{k=1}^{K} \nabla F_i(\boldsymbol{x}_{r,k}^i, \xi_{r,k}^i) \right\|^2$$

$$\leq \eta_c^2 \mathbb{E}_r \left\| \frac{1}{n} \sum_{i \in \mathcal{S}_r^a} \sum_{k=1}^{K} \nabla F_i(\boldsymbol{x}_{r,k}^i) \right\|^2 + \frac{K\eta_c^2}{n}\sigma^2$$

$$= \eta_c^2 \mathbb{E}_r \left\| \frac{1}{n} \sum_{i \in \mathcal{S}_r^a} \sum_{k=1}^{K} \left[\nabla F_i(\boldsymbol{x}_{r,k}^i) - \nabla F_i(\boldsymbol{x}_r) + \nabla F_i(\boldsymbol{x}_r)\right] \right\|^2 + \frac{K\eta_c^2}{n}\sigma^2$$

$$\leq 2\eta_c^2 \mathbb{E}_r \left\| \frac{1}{n} \sum_{i \in \mathcal{S}_r^a} \sum_{k=1}^{K} \left[\nabla F_i(\boldsymbol{x}_{r,k}^i) - \nabla F_i(\boldsymbol{x}_r)\right] \right\|^2 + 2\eta_c^2 \mathbb{E}_r \left\| \frac{1}{n} \sum_{i \in \mathcal{S}_r^a} \sum_{k=1}^{K} \nabla F_i(\boldsymbol{x}_r) \right\|^2 + \frac{K\eta_c^2}{n}\sigma^2$$

$$\leq \frac{2K\eta_c^2}{n} \mathbb{E}_r \sum_{i \in \mathcal{S}_r^a} \sum_{k=1}^{K} \left\| \nabla F_i(\boldsymbol{x}_{r,k}^i) - \nabla F_i(\boldsymbol{x}_r) \right\|^2$$

$$+ 2K^2\eta_c^2 \mathbb{E}_r \left\| \frac{1}{n} \sum_{i \in \mathcal{S}_r^a} \left[\nabla F_i(\boldsymbol{x}_r) - \nabla F_i(\boldsymbol{x}^*) + F_i(\boldsymbol{x}^*)\right] \right\|^2 + \frac{K\eta_c^2}{n}\sigma^2$$

$$= \frac{2L^2K\eta_c^2}{n} \mathbb{E}_r \sum_{i \in \mathcal{S}_r^a} \sum_{k=1}^{K} \left\| \boldsymbol{x}_{r,k}^i - \boldsymbol{x}_r \right\|^2 + \frac{4K^2\eta_c^2}{n} \mathbb{E}_r \sum_{i \in \mathcal{S}_r^a} \left\| \nabla F_i(\boldsymbol{x}_r) - \nabla F_i(\boldsymbol{x}^*) \right\|^2$$

$$+ \frac{4K^2\eta_c^2}{n} \mathbb{E}_r \sum_{i \in \mathcal{S}_r^a} \left\| \nabla F_i(\boldsymbol{x}^*) \right\|^2 + \frac{K\eta_c^2}{n}\sigma^2$$

$$\leq 2L^2K^2\eta_c^2 \left(5K\eta_c^2(\sigma^2 + 6K\sigma_G^2) + 30L^2K^2\eta_c^2\|\boldsymbol{x}_r - \boldsymbol{x}^*\|^2\right) + 4L^2K^2\eta_c^2 \|\boldsymbol{x}_r - \boldsymbol{x}^*\|^2$$

$$+ 4K^2\eta_c^2\sigma_G^2 + \frac{K\eta_c^2}{n}\sigma^2 \tag{36}$$

$$= (4L^2K^2\eta_c^2 + 60L^4K^4\eta_c^4) \|\boldsymbol{x}_r - \boldsymbol{x}^*\|^2 + (4K^2\eta_c^2 + 60L^2K^4\eta_c^4)\sigma_G^2$$

$$+ \left(\frac{K\eta_c^2}{n} + 10L^2K^3\eta_c^4\right)\sigma^2 \tag{37}$$

Plugging (35) and (37) into (31) and absorbing high-order terms into low-order terms, we get

$$\mathbb{E}_r^a\|\boldsymbol{x}_{r+1} - \boldsymbol{x}^*\|^2 \leq \|\boldsymbol{x}_r - \boldsymbol{x}^*\|^2 + \left(\frac{K\eta_c}{2\epsilon} + 60LK^3\eta_c^3\right)\sigma_G^2$$

$$+ \left(2K\eta_c\epsilon + 60L^3K^3\eta_c^3 - \frac{3\mu K\eta_c}{2}\right)\|\boldsymbol{x}_r - \boldsymbol{x}^*\|^2 + 10LK^2\eta_c^3\sigma^2$$

$$+ (4L^2K^2\eta_c^2 + 60L^4K^4\eta_c^4)\|\boldsymbol{x}_r - \boldsymbol{x}^*\|^2 + (4K^2\eta_c^2 + 60L^2K^4\eta_c^4)\sigma_G^2$$

$$+ \left(\frac{K\eta_c^2}{n} + 10L^2K^3\eta_c^4\right)\sigma^2$$

$$= \left(1 - \frac{3\mu K\eta_c}{2} + 2K\eta_c\epsilon + 4L^2K^2\eta_c^2 + 60L^3K^3\eta_c^3 + 60L^4K^4\eta_c^4\right)\|\boldsymbol{x}_r - \boldsymbol{x}^*\|^2$$

$$+ \left(\frac{K\eta_c}{2\epsilon} + 4K^2\eta_c^2 + 60LK^3\eta_c^3 + 60L^2K^4\eta_c^4\right)\sigma_G^2$$

$$+ \left(\frac{K\eta_c^2}{n} + 10LK^2\eta_c^3 + 10L^2K^3\eta_c^4\right)\sigma^2$$

$$\leq \left(1 - \frac{3\mu K\eta_c}{2} + 2K\eta_c\epsilon + 8L^2K^2\eta_c^2\right)\|\boldsymbol{x}_r - \boldsymbol{x}^*\|^2 + \left(\frac{K\eta_c}{2\epsilon} + 8K^2\eta_c^2\right)\sigma_G^2$$

$$+ \frac{2K\eta_c^2}{n}\sigma^2, \tag{38}$$

where the last inequality above follows the condition $\eta_c \leq \frac{1}{20nLK}$.

By (30) and (38), we establish:

$$\mathbb{E}_r\|\boldsymbol{x}_{r+1} - \boldsymbol{x}^*\|^2 = q\mathbb{E}_r^u\|\boldsymbol{x}_{r+1} - \boldsymbol{x}^*\|^2 + (1-q)\mathbb{E}_r^a\|\boldsymbol{x}_{r+1} - \boldsymbol{x}^*\|^2$$

$$\leq \left(1 - \frac{3\mu K\eta_c}{2} + (1-q)2K\eta_c\epsilon + 8L^2K^2\eta_c^2\right)\|\boldsymbol{x}_r - \boldsymbol{x}^*\|^2$$

$$+ \left((1-q)\frac{K\eta_c}{2\epsilon} + 8K^2\eta_c^2\right)\sigma_G^2 + \left(\frac{4K(qn + (1-q)m)\eta_c^2}{mn}\right)\sigma^2$$

$$\leq (1 - \mu K\eta_c)\|\boldsymbol{x}_r - \boldsymbol{x}^*\|^2 + \frac{(1-q)}{2}K^2\eta_c^2 + 8K^2\eta_c^2\sigma_G^2$$

$$+ \frac{2(qn + (1-q)m)}{mn}K\eta_c^2\sigma^2, \tag{39}$$

where the last inequality follows $\epsilon = \frac{\sigma_G^2}{K\eta_c}$, the condition on $\eta_c$ that we got before, and the condition $q \geq 1 - \left(\frac{\mu K\eta_c - 16L^2K^2\eta_c^2}{4\sigma_G^2}\right)$.

Applying (39) $R$ rounds recursively and denoting as $\kappa := \|\boldsymbol{x}_0 - \boldsymbol{x}^*\|^2$, we obtain

$$\mathbb{E}\|\boldsymbol{x}_R - \boldsymbol{x}^*\|^2 \leq (1 - \mu K\eta_c)^R\kappa + \frac{(1-q)}{2\mu}K\eta_c + \frac{8}{\mu}K\eta_c\sigma_G^2 + \frac{2(qn + (1-q)m)}{mn\mu}\eta_c\sigma^2$$

$$= \exp\left(R\ln(1 - \mu K\eta_c)\right)\kappa + \frac{(1-q)}{2\mu}K\eta_c + \frac{8}{\mu}K\eta_c\sigma_G^2 + \frac{2(qn + (1-q)m)}{mn\mu}\eta_c\sigma^2$$

$$\leq \exp\left(-\mu KR\eta_c\right)\kappa + \frac{(1-q)}{2\mu}K\eta_c + \frac{8}{\mu}K\eta_c\sigma_G^2 + \frac{2(qn + (1-q)m)}{mn\mu}\eta_c\sigma^2$$

$$= \exp\left(-\mu KR\eta_c\right)\kappa + \underbrace{\left(\frac{(1-q)}{2\mu}K + \frac{8}{\mu}K\sigma_G^2 + \frac{2(qn + (1-q)m)}{mn\mu}\sigma^2\right)}_{a}\eta_c \tag{40}$$

Choosing $\eta_c = \min\left\{\frac{\log(\max(\kappa\mu KR/a, 1))}{\mu KR}, \frac{1}{20mLK}, \frac{1}{20nLK}\right\}$, FAST can achieve the convergence rate of $\widetilde{\mathcal{O}}(1/R)$.

Assume that $m = n$, $\eta_c \leq \frac{1}{20mLK}$ and $R \geq 20mL$.

$$\mathbb{E}\|\boldsymbol{x}_R - \boldsymbol{x}^*\|^2 = \widetilde{\mathcal{O}}\left(\exp(-\frac{\mu}{20mL}R) + \frac{B^2}{\mu mKR}\right)$$

$$= \widetilde{\mathcal{O}}\left(\exp(-\frac{\mu}{20mL}R)\right) + \widetilde{\mathcal{O}}\left(\frac{(1-q)}{\mu R}\right) + \widetilde{\mathcal{O}}\left(\frac{1}{\mu R}\sigma_G^2\right) + \widetilde{\mathcal{O}}\left(\frac{1}{\mu mKR}\sigma^2\right),$$

where $B^2 = \frac{(1-q)mK}{2} + 8mK\sigma_G^2 + 2\sigma^2$. ∎

## B  ADDITIONAL EXPERIMENT DETAILS

### B.1  EXPERIMENT SETUP

#### B.1.1  MODELS

We design different kinds of models for different tasks and datasets. All details about the models we used are provided in our source codes.

a) Fashion-MNIST: The CNN model for Fashion-MNIST comprises two convolutional layers, introducing non-linearity through ReLU activation and incorporating max-pooling to reduce spatial dimensions. Additionally, it includes two fully connected layers to map features extracted by the convolutional layers to the final output categories.

b) CIFAR-10: The CNN model for CIFAR-10 consists of three convolutional layers followed by ReLU activation and max-pooling, and two fully connected layers.

c) Shakespeare: The Char-LSTM for Shakespeare incorporates an embedding layer to convert character indices into dense vectors, a two-layer LSTM, a dropout layer for regularization and a final linear layer generating prediction.

#### B.1.2  DATA DISTRIBUTION HETEROGENEITY

We use Dirichlet distribution to simulate the Non-IID data on Fashion-MNIST and CIFAR-10 datasets. The Shakespeare dataset is naturally Non-IID, so there is no demand to use Dirichlet distribution on it, and we can directly distribute each character's lines to each client to achieve Non-IID data distribution. Assuming that there are 30 clients, we set different $\alpha$ (the concentration parameter of Dirichlet distribution) to control the degree of Non-IID in data distribution. For a clear display, we visualize it in Figure 1. In each figure below, each color represents a class of label, meaning that there are 10 kinds of colors in the figures below.

When $\alpha = 0.05$, which is highly Non-IID, some clients only possess data with approximate $2 \sim 4$ types of labels, and data distribution varies greatly among clients, which is shown in Figure 1 (a). When $\alpha = 1$, each client owns data with all labels, and data distributions on every client are less Non-IID, as shown in Figure 1 (b).

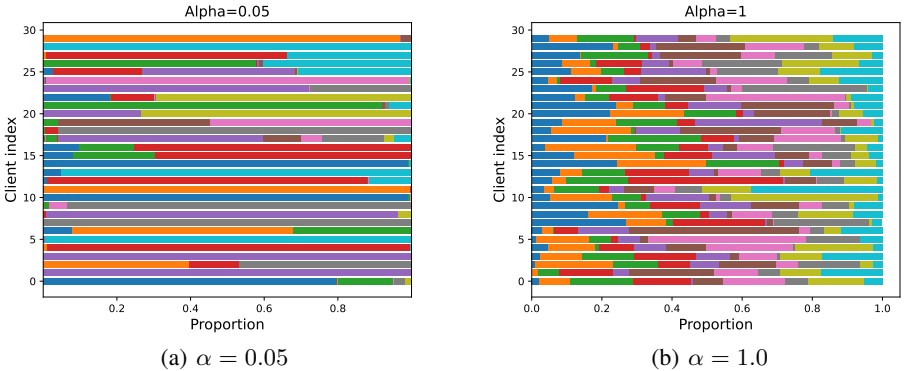

(a) $\alpha = 0.05$          (b) $\alpha = 1.0$

Figure 1: Dirichlet distribution with balanced data in each client

#### B.1.3  SIMULATION OF ARBITRARY CLIENT PARTICIPATION

In our experiments, we employ Beta, Gamma and Weibull distribution to simulate arbitrary client participation patterns in our FL system. To create highly heterogeneous client participation patterns, we select certain parameters of these distributions in our experiments, which will be introduced in

details in the following description. In addition, we show the participating situation of 100 clients in 1000 rounds under our parameter settings in Figure 2.

a) Beta distribution.

The probability density function of Beta distribution is $f(x; \alpha, \beta) = \frac{x^{\alpha-1}(1-x)^{\beta-1}}{B(\alpha, \beta)}$. It is characterized by two parameters, $\alpha$ and $\beta$, where $\alpha$ represents the number of successes in choosing and $\beta$ represents the number of failures in choosing, so we can adjust $\alpha$ and $\beta$ to control the preference of sampling. In our experiment, we set $\alpha = 1$ and $\beta = 10$.

b) Gamma distribution.

The probability density function of Gamma distribution is $f(x; k, \theta) = x^{k-1} \frac{e^{-x/\theta}}{\theta^k \Gamma(k)}$, where $k$ is shape parameter, $\theta$ is scale parameter, and $\Gamma$ is the Gamma function. In our experiment, we set $k = 10$ and $\theta = 0.01$.

c) Weibull distribution.

The probability density function of the Weibull distribution is

$$f(x; \lambda, k) = \begin{cases} \frac{\lambda}{k}\left(\frac{x}{\lambda}\right)^{k-1} e^{-(\frac{x}{\lambda})^k} & , \ x \geq 0 \\ 0 & , \ x < 0 \end{cases}$$

where $k$ is shape parameter, $\lambda$ is scale parameter. In our experiment, we set $k = 10$ and $\lambda = 1$.

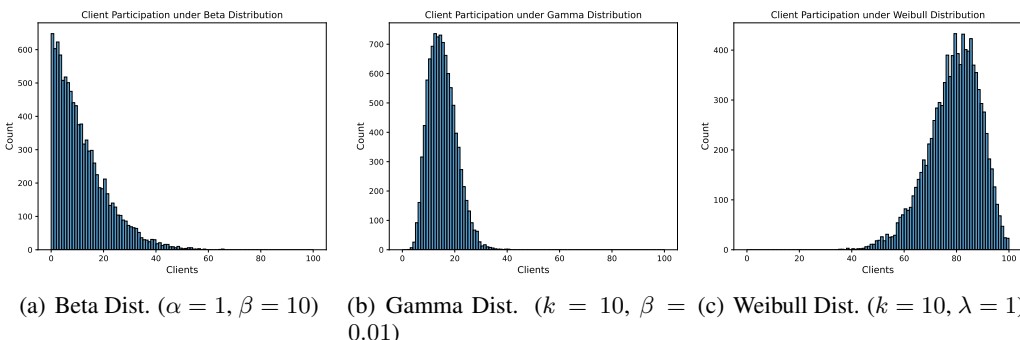

(a) Beta Dist. ($\alpha = 1, \beta = 10$) (b) Gamma Dist. ($k = 10, \beta = 0.01$) (c) Weibull Dist. ($k = 10, \lambda = 1$)

Figure 2: Client participation under Beta, Gamma and Weibull distribution

## B.2 Additional Experiment Results

### B.2.1 Additional Results of Impact of Arbitrary Client Participation

In our main paper, we have demonstrated that arbitrary client participation can degrade the performance of FL when using the classic FedAvg algorithm. To further verify the ubiquity of adverse effects of arbitrary client participation, we conducted extra related experiments to verify the impact of arbitrary client participation on other FL optimization algorithms. Table 4 shows the performance degradation resulting from arbitrary client participation when the FedProx Li et al. (2020) algorithm is used. Under the less Non-IID circumstances (e.g., $\alpha \geq 0.5$), the accuracy of the four types of participation is concentrated on around $90\%$. However, in highly heterogeneous situations, compared to uniform participation, the accuracy of the three types of arbitrary participation patterns suffers from a more severe decline. More specifically, when $\alpha = 0.05$, uniform participation suffers from $8\%$ accuracy degradation. However, the accuracy of Beta, Gamma and Weibull on the Fashion-MNIST dataset falls by around $15\%$, $16\%$ and $11\%$, respectively, which are more severe than the one under uniform client participation. In another heterogeneous situation with $\alpha = 0.1$, the accuracy of Beta, Gamma and Weibull on the CIFAR-10 dataset decreases by about $4\%$, $14\%$ and $5\%$ separately, but the accuracy of uniform client participation is just roughly reduced by $2\%$.

### B.2.2 ADDITIONAL RESULTS ABOUT REGULAR FAST

We demonstrate additional experiment results to verify the effectiveness of FAST and adaptive FAST.

Figure 3 demonstrates the performance of FAST+FedAvg. The corresponding numerical results have been shown in Table 3. All the performance of using FAST is beyond the baseline ($q = 0$), and lower than or even almost equal to the completely uniform participation ($q = 1$) sometimes.

Table 6 shows the performance of the FAST+FedProx Li et al. (2020) algorithm.

Table 7 shows the performance of the FAST+FedAvgM Hsu et al. (2019) algorithm.

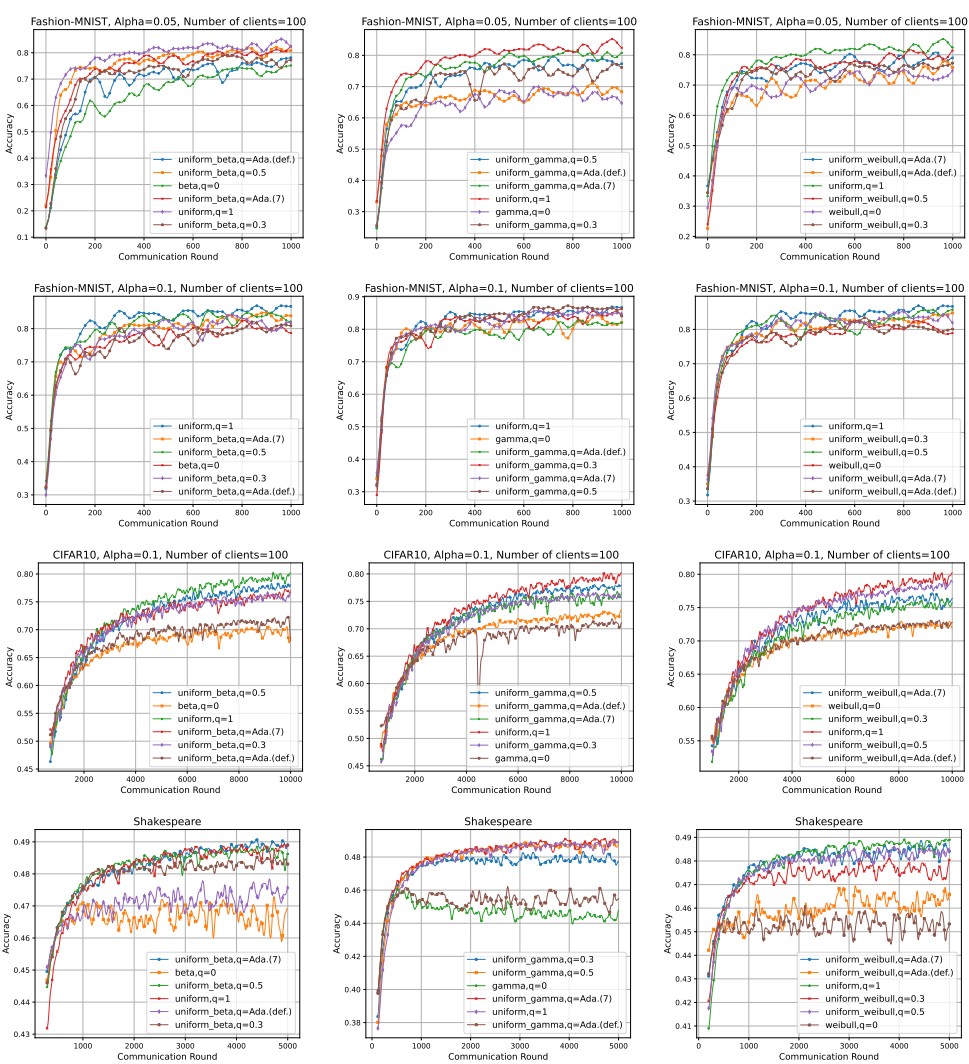

Figure 3: Experiment results of FAST+FedAvg on Fashion-MNIST & CIFAR-10 & Shakespeare

### B.2.3 ADDITIONAL RESULTS ABOUT ADAPTIVE FAST

To select possibly optimal $\lambda$, we test the performance of adaptive FAST using different $\lambda$ on the Fashion-MNIST dataset. Generally, the proportion of arbitrary client participation decreases as the increase in $\lambda$. Considering the accuracy and the proportion of arbitrary client participation, we empirically find that a big $\lambda$ (i.e., 6, 7) can result in a better balance between accuracy and the

Table 6: Experiment results of FAST+FedProx.

| Participation | q | Fashion-MNIST | | | | CIFAR-10 | | Shakespeare | |
| | | α=0.05 | | α=0.1 | | α=0.1 | | α =N/A | |
| | | Test Accuracy | Ratio | Test Accuracy | Ratio | Test Accuracy | Ratio | Test Accuracy | Ratio |
|---|---|---|---|---|---|---|---|---|---|
| Uniform (FedProx) | 1 | 83.48%±3.9 | 0% | 86.67%±0.5 | 0% | 79.53%±0.5 | 0% | 48.51%±0.6 | 0% |
| | Ada.(7) | 81.59%±3.9 | 59.7% | 85.21%±2.4 | 72.5% | 76.35%±1.4 | 65.0% | 48.66%±0.1 | 62.5% |
| | Ada.(def.) | 78.48%±1.6 | 88.1% | 83.65%±1.6 | 93.5% | 74.24%±1.3 | 96.5% | 47.50%±0.3 | 94.3% |
| | 0.5 | 78.78%±3.9 | 48.8% | 85.75%±2.2 | 50.7% | 76.96%±1.4 | 47.9% | 48.35%±0.4 | 50.5% |
| Beta (FAST) | 0.4 | 81.21%±2.2 | 60.5% | 84.94%±3.7 | 60.2% | 78.13%±0.6 | 58.5% | 48.61%±0.1 | 59.3% |
| | 0.3 | 78.64%±1.6 | 70.4% | 83.49%±2.2 | 68.9% | 75.77%±2.9 | 69.9% | 48.23%±0.4 | 70.2% |
| | 0.2 | 76.48%±2.3 | 81.9% | 80.59%±4.2 | 79.1% | 75.71%±0.9 | 80.3% | 48.49%±0.2 | 79.8% |
| | 0.1 | 75.01%±4.5 | 89.6% | 83.50%±1.4 | 89.5% | 72.80%±2.9 | 89.5% | 47.78%±0.3 | 90.5% |
| Beta (FedProx) | 0 | 74.74%±0.8 | 100% | 77.88%±4.7 | 100% | 72.76%±1.0 | 100% | 46.98%±0.4 | 100% |
| | Ada.(7) | 83.83%±2.3 | 61.6% | 85.34%±2.5 | 73.6% | 77.67%±2.3 | 63.4% | 47.96%±0.4 | 50.4% |
| | Ada.(def.) | 79.84%±2.4 | 93.8% | 85.98%±1.6 | 94.6% | 77.30%±0.7 | 95.4% | 44.36%±0.3 | 93.9% |
| | 0.5 | 84.93%±1.7 | 50.4% | 87.00%±1.6 | 48.2% | 77.34%±1.2 | 49.6% | 48.89%±0.1 | 49.5% |
| Gamma (FAST) | 0.4 | 81.43%±2.5 | 56.6% | 87.27%±1.1 | 58.1% | 77.51%±1.0 | 60.1% | 48.05%±0.2 | 58.9% |
| | 0.3 | 77.21%±0.9 | 67.8% | 85.84%±2.5 | 73.8% | 76.99%±2.2 | 70.7% | 46.98%±0.8 | 70.8% |
| | 0.2 | 81.33%±1.5 | 80.6% | 86.74%±1.8 | 79.6% | 77.50%±1.5 | 79.7% | 46.25%±0.8 | 80.6% |
| | 0.1 | 78.59%±2.4 | 90.2% | 86.15%±1.2 | 89.4% | 74.77%±0.6 | 89.7% | 44.48%±0.4 | 91.3% |
| Gamma (FedProx) | 0 | 75.59%±5.5 | 100% | 84.49%±1.9 | 100% | 65.94%±1.3 | 100% | 43.34%±0.5 | 100% |
| | Ada.(7) | 84.46%±1.7 | 55.8% | 86.01%±1.0 | 64.4% | 77.24%±1.4 | 69.3% | 48.70%±0.3 | 55.5% |
| | Ada.(def.) | 83.66%±2.6 | 92.9% | 84.11%±1.1 | 95.4% | 76.34%±0.8 | 96.4% | 47.88%±0.6 | 92.6% |
| | 0.5 | 82.39%±2.2 | 48.0% | 86.53%±1.6 | 48.2% | 78.30%±0.6 | 50.2% | 48.69%±0.2 | 50.3% |
| Weibull (FAST) | 0.4 | 79.16%±4.2 | 61.4% | 85.76%±1.9 | 59.1% | 76.64%±2.6 | 59.8% | 48.57%±0.2 | 60.3% |
| | 0.3 | 80.60%±3.5 | 67.2% | 85.72%±1.7 | 70.8% | 77.00%±1.1 | 70.2% | 48.37%±0.4 | 70.1% |
| | 0.2 | 80.73%±2.4 | 80.6% | 85.26%±2.3 | 81.4% | 76.78%±0.7 | 79.9% | 48.14%±0.2 | 80.6% |
| | 0.1 | 79.73%±2.8 | 91.5% | 84.65%±1.7 | 90.4% | 76.37%±1.3 | 89.6% | 48.23%±0.2 | 89.9% |
| Weibull (FedProx) | 0 | 79.55%±2.0 | 100% | 83.66%±2.3 | 100% | 72.94%±1.2 | 100% | 47.64%±0.8 | 100% |

proportion of arbitrary client participation than a small one. The results are shown in Table 8 and Table 5.

Figure 4 shows the distribution of $q$ and how $q$ in adaptive FAST varies during the entire training process. We find that $\lambda$ is related to the fluctuation range of $q$. Generally, the larger $\lambda$ is, the more drastic the change in $q$, thus giving rise to a higher proportion of uniform participation (*Ratio*). Additionally, when $\lambda = 1$, the values of $q$ are primarily concentrated on $0 \sim 0.2$, resulting from that a small parameter $\lambda$ limits the fluctuation of $q$. In contrast with a small $\lambda$, a large $\lambda$ (i.e., 7) leads to a more uniform distribution of $q$.

Table 7: Experiment results of FAST+FedAvgM.

| Participation | q | Fashion-MNIST | | | | CIFAR-10 | | Shakespeare | |
| | | α=0.05 | | α=0.1 | | α=0.1 | | α =N/A | |
| | | Test Accuracy | Ratio | Test Accuracy | Ratio | Test Accuracy | Ratio | Test Accuracy | Ratio |
|---|---|---|---|---|---|---|---|---|---|
| Uniform (FedAvgM) | 1 | 84.23%±1.3 | 0% | 87.21%±2.0 | 0% | 77.74%±2.3 | 0% | 48.66%±0.2 | 0% |
| | Ada.(7) | 83.86%±1.4 | 61.0% | 85.27%±3.1 | 65.9% | 76.92%±1.6 | 66.7% | 48.77%±0.1 | 60.2% |
| | Ada.(def.) | 80.38%±1.5 | 93.0% | 83.40%±1.5 | 93.7% | 73.30%±0.7 | 96.8% | 47.69%±0.3 | 94.3% |
| | 0.5 | 83.52%±2.3 | 50.1% | 86.94%±1.7 | 51.5% | 77.61%±0.8 | 49.2% | 48.79%±0.5 | 50.3% |
| Beta (FAST) | 0.4 | 83.51%±2.5 | 58.9% | 84.97%±2.5 | 58.1% | 76.26%±1.6 | 59.9% | 48.41%±0.2 | 60.1% |
| | 0.3 | 83.54%±2.1 | 68.4% | 83.09%±4.8 | 69.4% | 76.37%±0.3 | 69.5% | 47.77%±1.3 | 70.2% |
| | 0.2 | 80.34%±1.3 | 77.9% | 84.20%±2.1 | 81.8% | 75.56%±1.5 | 79.8% | 48.24%±0.2 | 79.2% |
| | 0.1 | 70.96%±5.5 | 92.3% | 82.29%±1.4 | 90.4% | 74.54%±1.4 | 90.4% | 47.74%±0.2 | 89.3% |
| Beta (FedAvgM) | 0 | 70.48%±2.6 | 100% | 82.18%±2.9 | 100% | 72.96%±1.0 | 100% | 46.86%±0.8 | 100% |
| | Ada.(7) | 78.14%±2.4 | 72.8% | 83.26%±0.9 | 83.9% | 76.21%±0.7 | 70.3% | 48.09%±0.5 | 51.6% |
| | Ada.(def.) | 74.30%±2.2 | 92.3% | 80.22%±1.6 | 95.9% | 70.61%±0.7 | 98.1% | 44.64%±1.2 | 93.4% |
| | 0.5 | 78.08%±3.4 | 50.4% | 86.39%±1.8 | 50.9% | 77.59%±1.8 | 49.9% | 48.31%±0.3 | 51.0% |
| Gamma (FAST) | 0.4 | 79.58%±2.6 | 58.8% | 83.88%±2.4 | 61.4% | 77.15%±1.1 | 60.1% | 48.18%±0.3 | 58.5% |
| | 0.3 | 77.39%±1.5 | 71.9% | 83.56%±3.4 | 70.5% | 74.69%±1.2 | 70.1% | 47.14%±0.4 | 70.7% |
| | 0.2 | 75.29%±2.1 | 80.1% | 82.31%±2.2 | 79.3% | 73.67%±1.2 | 79.7% | 46.70%±0.4 | 79.3% |
| | 0.1 | 69.63%±3.1 | 92.2% | 81.08%±1.3 | 90.6% | 71.88%±0.7 | 89.9% | 44.33%±0.3 | 90.2% |
| Gamma (FedAvgM) | 0 | 65.76%±3.6 | 100% | 80.01%±1.5 | 100% | 67.40%±0.9 | 100% | 42.65%±0.4 | 100% |
| | Ada.(7) | 83.50%±3.1 | 62.8% | 85.12%±2.8 | 68.3% | 77.45%±0.8 | 67.5% | 48.90%±0.3 | 52.8% |
| | Ada.(def.) | 83.91%±1.0 | 94.2% | 83.32%±2.0 | 92.8% | 75.23%±1.2 | 96.5% | 48.00%±0.3 | 92.4% |
| | 0.5 | 83.16%±3.2 | 51.7% | 86.29%±1.4 | 50.9% | 78.17%±1.1 | 49.9% | 48.69%±0.2 | 51.1% |
| Weibull (FAST) | 0.4 | 81.93%±1.8 | 59.0% | 84.47%±2.1 | 62.3% | 77.86%±1.2 | 59.7% | 48.56%±0.5 | 60.6% |
| | 0.3 | 83.22%±3.1 | 71.4% | 83.54%±2.2 | 70.6% | 76.61%±0.9 | 70.0% | 48.64%±0.2 | 69.5% |
| | 0.2 | 82.67%±2.2 | 80.6% | 83.19%±1.7 | 79.3% | 76.71%±0.6 | 79.4% | 48.36%±0.2 | 79.3% |
| | 0.1 | 82.90%±3.2 | 89.1% | 83.38%±1.6 | 90.3% | 75.78%±0.9 | 90.2% | 48.38%±0.2 | 89.7% |
| Weibull (FedAvgM) | 0 | 75.97%±4.8 | 100% | 80.79%±2.8 | 100% | 74.38%±1.5 | 100% | 47.73%±0.3 | 100% |

Table 8: Performance comparison of different $\lambda$ for adaptive FAST+FedAvg with $\alpha = 0.05$.

| $\lambda$ | 1 (def.) | 2 | 3 | 4 | 5 | 6 | 7 | 8 | 9 |
|---|---|---|---|---|---|---|---|---|---|
| Beta (FAST) | 77.93%±0.7 | 76.15%±4.2 | 76.87%±3.8 | 79.95%±3.4 | 80.52%±3.4 | 78.69%±3.4 | 80.92%±3.1 | 79.65%±2.9 | 81.13%±1.6 |
| Ratio | 88.5% | 87.4% | 87.1% | 78.8% | 74.7% | 68.0% | 60.3% | 61.7% | 66.6% |
| Gamma (FAST) | 71.48%±4.5 | 78.95%±2.3 | 77.50%±4.7 | 80.08%±0.8 | 81.98%±1.1 | 81.53%±2.8 | 79.95%±4.9 | 82.07%±1.7 | 80.44%±3.7 |
| Ratio | 91.8% | 82.3% | 77.6% | 67.3% | 56.7% | 53.8% | 59.3% | 50.6% | 52.9% |
| Weibull (FAST) | 77.14%±2.7 | 73.52%±4.0 | 74.79%±4.0 | 79.50%±3.3 | 78.26%±4.9 | 80.04%±2.1 | 77.89%±3.3 | 80.87%±2.6 | 74.52%±8.9 |
| Ratio | 90.4% | 83.8% | 72.5% | 66.1% | 61.6% | 58.6% | 59.5% | 60.6% | 59.1% |

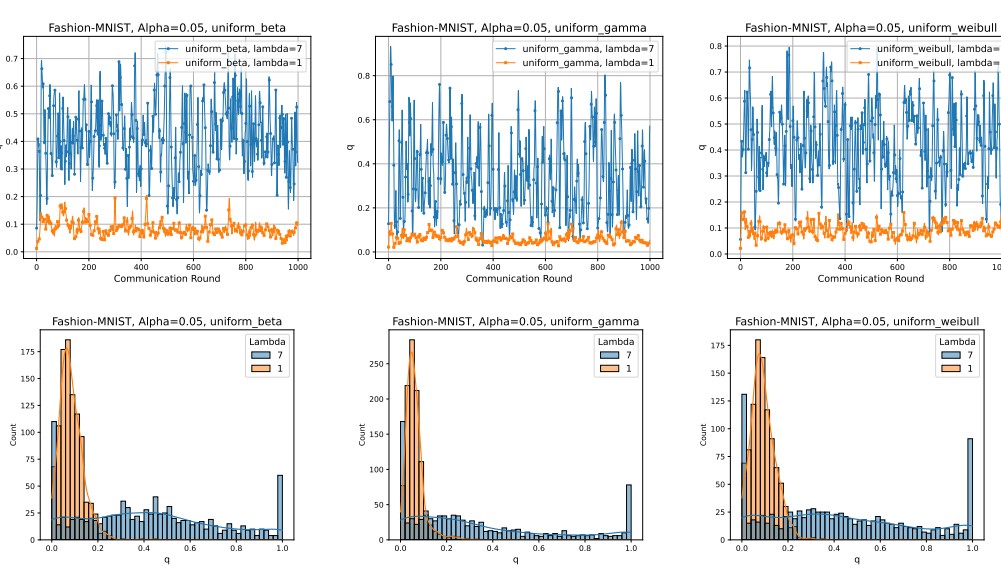

Figure 4: The fluctuation and distribution of $q$ using different $\lambda$ in adaptive FAST+FedAvg.

