# OpenReview forum: "FAST: Federated Average with Snapshot Unleashes Arbitrary Client Participation"
_ICLR.cc/2025/Conference — Submitted to ICLR 2025_

### Official Review · Reviewer_iE3h · 2024-11-01

**Soundness:** 2
**Presentation:** 2
**Contribution:** 2
**Rating:** 3
**Confidence:** 3

**Summary:**

This paper investigates the impact of arbitrary client participation (ACP) in Federated Learning (FL) and proposes a solution called Federated Average with Snapshot (FAST). FAST enables FL systems to handle unpredictable client participation by allowing clients to take periodic snapshots, which helps stabilize learning. The authors show that FAST maintains effective convergence rates, even in challenging conditions with non-convex and heterogeneous data. The paper also introduces an adaptive strategy to dynamically adjust snapshot frequency, demonstrating that FAST improves FL performance under ACP.

**Strengths:**

This paper proposed a new strategy to tackle partial client participation in FL.

**Weaknesses:**

I think this algorithm can't beat the existing algorithm such as FedAU. Their analysis doesn't rely on bounded data heterogeneity assumption. They don't force all the clients to participate in some rounds.

This strategy is hard to implement. How could you know $q$? Additionally, this analysis is not tight when it reduces to an IID case. When it comes to the IID case, q doesn't reduce to 0 according to their analysis.

Although they discuss q in the manuscript, that's not enough. The definition of G is important which was missed in the main text. You should give more discussion of q so that the readers can see how tight is your bound, and how effective is your algorithm.

There are a lot of algorithms proposed for handling partial client participation, all of these algorithm are missed in the comparison in the experiments.

**Questions:**

Please refer to the weakness. I am happy to discuss. But currently, I am negative about this work.

---

### Official Review · Reviewer_fABB · 2024-11-02

**Soundness:** 2
**Presentation:** 3
**Contribution:** 2
**Rating:** 5
**Confidence:** 4

**Summary:**

This paper presents FAST, a federated learning framework aimed at addressing the ACP issue. ACP refers to the unpredictable and heterogeneous client participation observed in FL. FAST incorporates periodic ``snapshot step’’ where uniform client participation is enforced, allowing the model to periodically capture a more representative distribution of client updates. The theoretical analysis supports that FAST achieves convergence rates comparable to those under ideal conditions of uniform participation.

**Strengths:**

+ The paper tackles ACP, a critical challenge in real-world FL scenarios where client availability varies significantly.

+ The snapshot steps provide a novel approach to balancing random client participation with periodic uniform participation, potentially reducing the impact of ACP.

+ The theoretical results demonstrate that FAST achieves desirable convergence rates under both non-convex and strongly convex settings, indicating robustness.

**Weaknesses:**

- While the snapshot steps aim to ensure periodic uniform participation, it may be challenging in practice to ensure that all relevant clients are available during these rounds. In real-world FL scenarios, client availability is often impacted by network limitations, resource constraints, and other unpredictable factors. If key clients are unavailable during snapshot steps, uniform participation may be incomplete, reducing the snapshots’ effectiveness in mitigating biases introduced by ACP.

- The adaptive mechanism for adjusting snapshot frequency based on single-round accuracy changes may be sensitive to noise, potentially leading to frequent or unnecessary adjustments. Additionally, it does not account for other factors, such as client dropout rates or long-term accuracy trends, which may reduce its robustness in highly dynamic environments.

- ACP in FL can lead to a ``forgetting problem’’, where the model gradually loses knowledge of certain client data distributions due to inconsistent client participation, especially in highly non-IID scenarios. Although FAST's snapshot mechanism could mitigate this by enforcing periodic uniform participation, it may not fully address forgetting because: (i) Snapshot steps account for only a small portion of training rounds. Between these steps, updates are based on arbitrary participation, which could lead to a bias toward data from more frequently participating clients, causing the model to overlook less frequently updated distributions. (ii) The adaptive mechanism in Algorithm 2, which relies on immediate accuracy changes, lacks sensitivity to the gradual forgetting that can emerge in non-IID data.

- The paper does not compare FAST with other adaptive participation methods, such as FedNova [1] or FedMA [2], which also address client participation variability under heterogeneous conditions.

[1] Jianyu Wang, et al., Tackling the objective inconsistency problem in heterogeneous federated optimization. In Proc. of NeurIPS, 2020.

[2] Hongyi Wang, et al., Federated learning with matched averaging. In Proc. of ICLR, 2020.

**Questions:**

- Given the challenges of achieving uniform participation in real-world FL scenarios, have methods like increased sampling size or multi-round snapshot accumulation been considered to improve client representation during snapshot steps?

- Have methods been considered to reduce fluctuations in accuracy, such as a smoothing mechanism (e.g., moving average), incorporating additional participation metrics, or adding constraints on snapshot frequency adjustments?

- Has the potential for model forgetting in highly non-IID scenarios been analyzed, especially in cases where certain client data may be underrepresented over rounds?

- Adding comparisons with other adaptive participation methods (such as FedNova [1] and FedMA [2]) could provide a more comprehensive evaluation of FAST’s performance under ACP.

[1] Jianyu Wang, et al., Tackling the objective inconsistency problem in heterogeneous federated optimization. In Proc. of NeurIPS, 2020.

[2] Hongyi Wang, et al., Federated learning with matched averaging. In Proc. of ICLR, 2020.

---

### Official Review · Reviewer_tKYL · 2024-11-02

**Soundness:** 2
**Presentation:** 2
**Contribution:** 2
**Rating:** 5
**Confidence:** 4

**Summary:**

This paper explores the impact of arbitrary client participation in federated learning (FL) and proposes a lightweight client participation mechanism, FAST, aided by snapshots, to improve FL performance in settings with arbitrary client participation. Experimental results validate the improvements achieved by FAST.

**Strengths:**

1. The paper is well written.
2. Clear comparison of convergence rate with other existing schemes.
3. Convergence analysis is provided.
4. The proposed algorithm FAST is easy to implemented.

**Weaknesses:**

1. The motivation of arbitrary client participation (ACP) is unclear. Existings clients sampling schemes typically assign different sampling probability to different clients based on their data heterogeneity or data distribution. The motivation of using Beta, Gamma or Weibull distribution is hard to understand.
2. There is no comparison between FAST and existing cleint sampling schemes [1-4].
3. The datasets used in the experiments are simple and old. Larger datasets such as cifar100 or imagenet should be considered.
4. Only CNN is used as the model architecture in the experiments. Performance of Fast on complicated model architectures remains unknown.


Reference:

[1] Cho Y J, Wang J, Joshi G. Client selection in federated learning: Convergence analysis and power-of-choice selection strategies

[2] Fraboni Y, Vidal R, Kameni L, et al. Clustered sampling: Low-variance and improved representativity for clients selection in federated learning

[3] Balakrishnan R, Li T, Zhou T, et al. Diverse client selection for federated learning via submodular maximization.

[4] Chen H, Vikalo H. Heterogeneity-Guided Client Sampling: Towards Fast and Efficient Non-IID Federated Learning

**Questions:**

1. What's the difference between uniformly random sampling and  arbitrarily random sampling? Random sampling is considered an inefficient sampling way for heterogeneous federated learning.
2. Does FAST has faster convergence rate than other schemes. They looks the same in the table 1.

---

### Official Review · Reviewer_KjF2 · 2024-11-04

**Soundness:** 2
**Presentation:** 2
**Contribution:** 2
**Rating:** 3
**Confidence:** 2

**Summary:**

This paper studies arbitrary client participation (ACP) in federated learning (FL). The authors propose a mechanism called Federated Average with Snapshot (FAST) to unleash almost ACP for FL. FAST periodically enforces uniform client participations at rounds for communications, and allows arbitrary random client participations with a fixed size of chosen clients at each round during each communication period. The authors also present the convergence analysis for FAST in non-convex and strongly-convex models. The authors conduct numerical experiments using Fashion-MNIST, CIFAR-10 and Shakespeare datasets, where the arbitrary random participations are simulated under Beta, Gamma and Weibull distributions, respectively.

**Strengths:**

This paper presents comprehensive comparisons between the proposed FAST and Fed-Avg (McMahan et al., 2017) under several settings of the random partial participations. This paper also contains complete proofs for both non-convex and strongly-convex models used in FAST for FL.

**Weaknesses:**

There seems to be potential confusing points in the description of client participations in the paper. See Questions for the details.

**Questions:**

In line 6 of Table 1 on page 2, the authors mention that FedAvg with the uniform participation in every round has the convergence rate of $O(1/\sqrt{mKR})$, where $R$ refers to number of rounds and $K$ refers to number of local steps taken in each client during each round. So, it seems that $m$ means the number of clients in the FL based on the phrase "uniform participation in every round". Then, in (1) on page 4, the authors use $M$ to denote the total number of clients, and $M$ is not used in the algorithm or the analysis later. On the other hand, line 4 of Algorithm 1 on page 6 says that the number of uniformly random clients chosen by the server in the snapshot round is $m$.

If $m$ is just $M$, then the description "the server enforces uniformly random clients" in line 4 of Algorithm 1 seems to be confusing.

If $m$ and $M$ are different notions, then it seems that the convergence rate of $O(1/\sqrt{mKR})$ for FedAvg is based on the uniformly random participation.

Additionally, the result of convergence rate of FAST $O(1/\sqrt{mKR})$ shown in line 7 of Table 1, the same as FedAvg with the uniform participation, seems to be from Corollary 2 on page 7. In Corollary 2, the authors suppose that $m=n$. If $m$ is just $M$, then $m=n=M$ means that FAST is trivially just like Fed-Avg based on line 4 and 6 of Algorithm 1. If $m$ and $M$ are different notions, then it seems that $m=n$ makes the difference between line 4 and 6 of Algorithm 1 negligible, since both cases need to randomly choose a subset of clients with the same size.


Could the authors provide some clarifications on the notions of $m$ and $M$ and the usage of $m=n$ in Corollary 2?

---

### Official Review · Reviewer_bauJ · 2024-11-05

**Soundness:** 3
**Presentation:** 3
**Contribution:** 3
**Rating:** 6
**Confidence:** 4

**Summary:**

The submission considers the arbitrary client participation in federated learning. It motivates the work by presenting an experimental exploration claiming that a uniform selection of clients in every round of federated averaging performs better in terms of generalization accuracy of the trained model compared to the clients selected with $\beta$, $\gamma$, and Weibull distributions. Thereby, it suggests that applying uniform sampling of clients intermittently between the rounds of arbitrary participation helps improve the training quality, which is also shown in the experiments.

The submission presents the convergence theory of the federated learning scheme where the number of rounds include both uniform and arbitrary client participation.

**Strengths:**

The work presents an interesting approach to addressing the degradation in training quality due to arbitrary client participation in Federated Learning. The motivation is good and the submission is well written and structured.

**Weaknesses:**

Unsurprisingly, when several clients do not participate in over a thousand rounds of training, the quality of the trained model will degrade. The experiments corroborate this fact. It becomes stark in cases such as gamma distribution where clients ranked ~40-100 almost never participate in federation. Nevertheless, the solution that uniform client selection will be warranted for several rounds leaves the requirement to ensure the same.

Here are a few rough edges:
1. In the experiments, ten clients participated in each round; it does not fully simulate the premise of $m\neq n$.  How about simulating something like when m>2n?

2. It is a bit unfair to not compare the results with similar related works. For example, it will be interesting to see how the presented scheme compares with FedAU: "A Lightweight Method for Tackling Unknown Participation Statistics in Federated Averaging
Shiqiang Wang, Mingyue Ji, 2024". In this context, it is unclear what is the motivation for including FedAvgM, Tsu and Brown, 2019.

3. Derivation of equation 24 is fair. However, it distracts a reader to refer to the lemmas from the cited papers too often.

**Questions:**

* Can you try simulating a combination of FAST with FedAU? It will be interesting to see how the performance behaves then.
* Though the final expressions in equations 24 and 38/39 look correct on the first reading, can the steps, such as telescoping sum, etc., be written more explicitly? Also, it will be helpful to write the lemmas from the cited papers for self-contained references.

---

### Meta-Review · Area_Chair_6A7B · 2024-12-15

**Metareview:**

The paper considers heterogeneous client participation in the federated learning setting. But as pointed out by multiple reviewers, the work has multiple limitations and is not yet ready for presentation.

**Additional Comments On Reviewer Discussion:**

The points raised by the reviewers were not addressed during the rebuttal period.

---

### Decision · Program_Chairs · 2025-01-22

Reject